# Sparse recurrent excitatory connectivity in the microcircuit of the adult mouse and human cortex

Stephanie C Seeman[1†], Luke Campagnola[1†], Pasha A Davoudian[1], Alex Hoggarth[1], Travis A Hage[1], Alice Bosma-Moody[1], Christopher A Baker[1], Jung Hoon Lee[1], Stefan Mihalas[1], Corinne Teeter[1], Andrew L Ko[2,3], Jeffrey G Ojemann[2,3], Ryder P Gwinn[4], Daniel L Silbergeld[3], Charles Cobbs[5], John Phillips[1], Ed Lein[1], Gabe Murphy[1], Christof Koch[1], Hongkui Zeng[1], Tim Jarsky[1]*

[1]Allen Institute for Brain Science, Seattle, United States; [2]Regional Epilepsy Center at Harborview Medical Center, Seattle, United States; [3]Department of Neurological Surgery, University of Washington School of Medicine, Seattle, United States; [4]Epilepsy Surgery and Functional Neurosurgery, Swedish Neuroscience Institute, Seattle, United States; [5]The Ben and Catherine Ivy Center for Advanced Brain Tumor Treatment, Swedish Neuroscience Institute, Seattle, United States

**Abstract** Generating a comprehensive description of cortical networks requires a large-scale, systematic approach. To that end, we have begun a pipeline project using multipatch electrophysiology, supplemented with two-photon optogenetics, to characterize connectivity and synaptic signaling between classes of neurons in adult mouse primary visual cortex (V1) and human cortex. We focus on producing results detailed enough for the generation of computational models and enabling comparison with future studies. Here, we report our examination of intralaminar connectivity within each of several classes of excitatory neurons. We find that connections are sparse but present among all excitatory cell classes and layers we sampled, and that most mouse synapses exhibited short-term depression with similar dynamics. Synaptic signaling between a subset of layer 2/3 neurons, however, exhibited facilitation. These results contribute to a body of evidence describing recurrent excitatory connectivity as a conserved feature of cortical microcircuits.
DOI: https://doi.org/10.7554/eLife.37349.001

*For correspondence:
timj@alleninstitute.org

[†]These authors contributed equally to this work

**Competing interests:** The authors declare that no competing interests exist.

## Introduction

Generating well-informed, testable hypotheses about how the cortex represents and processes information requires experimental efforts to characterize the connectivity and dynamics of cortical circuit elements as well as efforts to build models that integrate results across studies (*Sejnowski et al., 1988*). Estimates of connectivity and synaptic properties vary widely between experiments due to differences in model organisms, experimental parameters, and analytic methods. This variability limits our ability to generate accurate, integrative computational models.

Addressing this problem requires standardized experimental methods and large-scale data collection in order to characterize synaptic connections between the large number of potential cell types (*Tasic et al., 2016*). Although it may be possible to infer part of these results based solely on anatomical constraints (*Markram et al., 2015*), evidence has shown that the rate of connectivity and properties of synaptic signals can depend on the identity of the pre- and postsynaptic neuron (*Reyes et al., 1998*; *Galarreta and Hestrin, 1998*; *Larsen and Sjöström, 2015*). To collect standardized data at scale, we have established a pipeline to characterize local, functional connectivity in the

**eLife digest** The outer sheet of brain tissue, the neocortex, is composed of circuits formed from trillions of connections among billions of neurons, of which there are about one hundred different neuron types. The scale and complexity of cortical circuitry pose experimental challenges, leading to an incomplete understanding of how cortical cell types are connected and the computations that take place at the connections.

About half of the cell types in the brain are excitatory, which means they can activate other cells. The cortex consists of several distinct layers of cells, within which excitatory cells cooperate to process the signals they receive from other cortical layers and brain areas. Using recordings of electrical activity arising from the connections between pairs of excitatory neurons, Seeman, Campagnola et al. measured the likelihood and strength of connectivity among related groups of excitatory cell types in slices of cortex taken from human and mouse brains.

The initial results confirm previous findings that individual layers of human cortex can have more and stronger excitatory connections than the same layers of mouse cortex. In most layers of mouse cortex, repeatedly activating the excitatory cells leads to progressively weaker responses. However, in the upper layers of mouse cortex, the opposite effect is sometimes seen: more excitatory activity causes the connections to generate stronger responses. By feeding these data into a computer model, Seeman, Campagnola et al. described and compared the activity of the groups of related excitatory cell types.

These results are the first of a new, large-scale project where findings can be integrated across experiments to gain a more detailed picture of cortical circuitry and computation. Neuroscientists will be able to use the results to build advanced computer models of cortical circuits. Such models will, for example, generate predictions for how the attributes of excitatory connectivity revealed by Seeman, Campagnola et al. influence how information is processed in the cortex. In so doing, the models will add to our understanding of how the human brain works both in health and in disease.

DOI: https://doi.org/10.7554/eLife.37349.002

adult mouse and human cortex. Initially, we seek to characterize connectivity among cell classes, that is, groups of related cell types (*Tasic et al., 2016*). Here, we report on the characteristics of local excitatory inputs among pyramidal neurons from within the same layer (recurrent connections) obtained during the pipeline's system integration test—an end-to-end test of the pipeline's hardware, software, and workflow carried out prior to initializing the pipeline.

Recurrent excitatory connectivity is thought to be important in behavior (*Evans et al., 2018*) and disease (*Jin et al., 2006*). It is a common feature in computational models of cortical working memory, receptive field shaping, attractor dynamics, and sequence storage (*Camperi and Wang, 1998*; *Olshausen and Field, 1996*; *Mongillo et al., 2008*; *Brunel, 2016*; *Pernice et al., 2018*). Empirical measurements of recurrent connectivity and synaptic properties are needed in order to constrain and validate these models. However, characterizing recurrent connectivity in a standardized, high-throughput manner is challenging because the synaptic connections can be sparse and weak (*Braitenberg and Schüz, 1998*; *Song et al., 2005*; *Lefort et al., 2009*). Furthermore, most measurements of recurrent connectivity have been performed in juvenile rodents, leading to a recent debate over the rate of connectivity in the adult cortex (*Biane et al., 2015*; *Barth et al., 2016*; *Jiang et al., 2016*).

The data reported here demonstrate that sparse recurrent connectivity is present among excitatory neurons in all layers of adult mouse and human cortex. Using a novel automated method for systematically estimating connectivity across experiments, we further demonstrate that different populations of adult mouse pyramidal neurons exhibit characteristic distance-dependent connectivity profiles and short-term dynamics. Finally, we quantify and compare differences in short-term dynamics with a mechanistic computational model.

## Results

We performed *in vitro* whole-cell recordings from up to eight excitatory neurons simultaneously. We probed 2836 putative connections in mouse V1 from excitatory cell classes defined by transgenic

**Table 1.** The number of connections probed and the number of connections used in subsequent analyses per the analysis flow diagram in *Figure 1—figure supplement 1C–G*.

For each column, the *Figure 1—figure supplement 1* letter indicates the end level in the analysis flow diagram while the main figure reference indicates *n* connections included in that figure. For example, the 'Strength' column indicates the number of connections for each type used to measure the strength (or amplitude) of the connection as shown in *Figure 1F*. The inclusion criteria for these connections can be followed in the diagram in *Figure 1—figure supplement 1E*. Similarly, these data are provided for kinetics (rise time and latency) and short-term plasticity (STP).

| Layer/ Cell Type | Total probed (*Figure 1—figure supplement 1C*) | Total connected (*Figure 1—figure supplement 1C*) | Total connection probability (%) | Strength (*Figure 1—figure supplement 1E, Figure 1F*) | Kinetics (*Figure 1—figure supplement 1F, Figure 1F*) | Connection probability (%) w/in 100 μm (Connected/probed, *Figure 1—figure supplement 1D, Figure 4A,C*) | STP (*Figure 1—figure supplement 1G, Figures 5 and 6*) |
|---|---|---|---|---|---|---|---|
| Mouse L2/3 | 180 | 15 | 8.3 | 12 | 9 | 13/130 (10.0) | 9 |
| Rorb | 315 | 20 | 6.3 | 13 | 13 | 18/247 (7.3) | 9 |
| Tlx3 | 1108 | 39 | 3.5 | 17 | 14 | 36/746 (4.8) | 5 |
| Sim1 | 783 | 55 | 7.0 | 18 | 18 | 41/527 (7.8) | 7 |
| Ntsr1 | 450 | 2 | 0.4 | 2 | 2 | 0/313 (0.0) | N/A |
| Human L2 | 132 | 22 | 16.7 | 18 | 18 | 13/69 (18.8) | N/A |
| Human L3 | 249 | 37 | 14.9 | 33 | 29 | 20/106 (18.9) | N/A |
| Human L4 | 123 | 4 | 3.3 | 2 | 2 | 1/51 (2.0) | N/A |
| Human L5 | 112 | 13 | 11.6 | 6 | 6 | 6/49 (12.2) | N/A |

DOI: https://doi.org/10.7554/eLife.37349.003

labeling, morphology, and cortical layer (*Tasic et al., 2016*). We further probed 616 putative connections in human frontal and temporal cortex from excitatory cell classes defined by morphology and cortical layer (*Table 1*). Recurrent connectivity was tested and observed in layer 2/3 through layer 6 of mouse primary visual cortex and layer 2 through layer 6 of the human cortex. To assess connectivity, trains of action potentials were evoked in each cell, one at a time, while recording synaptic responses in all other cells. Connections were identified by the presence of excitatory postsynaptic potentials (EPSPs) evoked with a short latency and low jitter following the presynaptic spike, consistent with monosynaptic connections (*Figure 1—figure supplement 1*). We encountered no examples of EPSPs eliciting spikes in any recorded pyramidal cells, further indicating that evoked polysynaptic activity should be rare in these experiments.

## Properties of intralaminar excitatory synaptic signaling in mouse cortex

Layer and projection-specific classes of excitatory neuron populations were identified either by *post-hoc* morphologic evaluation in layer 2/3 (animals n = 11) or transgenic labelling to target layers 4–6 (layer 4: Rorb (n = 28), layer 5: Tlx3 (n = 57), Sim1 (n = 20), layer 6: Ntsr1 (n = 13); *Figure 1A*). Layer 5 recordings were subdivided into subcortical projecting cells (Sim1; http://connectivity.brain-map.org) or corticocortical projecting cells (Tlx3; *Kim et al., 2015*). In layer 6, only the subcortically projecting cells were targeted (Ntsr1; *Vélez-Fort et al., 2014*). We probed 2836 potential connections (layer 2/3: 180, Rorb: 315, Tlx3: 1108, Sim1: 783, Ntsr1: 450) across these excitatory populations in mouse cortex (*Table 1*). Connections were detected between 131 putative pre- and post-synaptic partners (layer 2/3: 15, Rorb: 20, Tlx3: 39, Sim1: 55, Ntsr1: 2; *Table 1*). For >75% of the recorded cells, we recovered a biocytin fill (*Figure 1A*) and for all cells we obtained an epifluorescent image stack (*Figure 1B*).

We first characterized the strength and kinetics in recurrent connections of each Cre-type and layer (*Figure 1*). To measure these features with minimal influence of STP, only the first response on each sweep (inter-trial interval (ITI) = 15 s) was included for this analysis. For each connection,

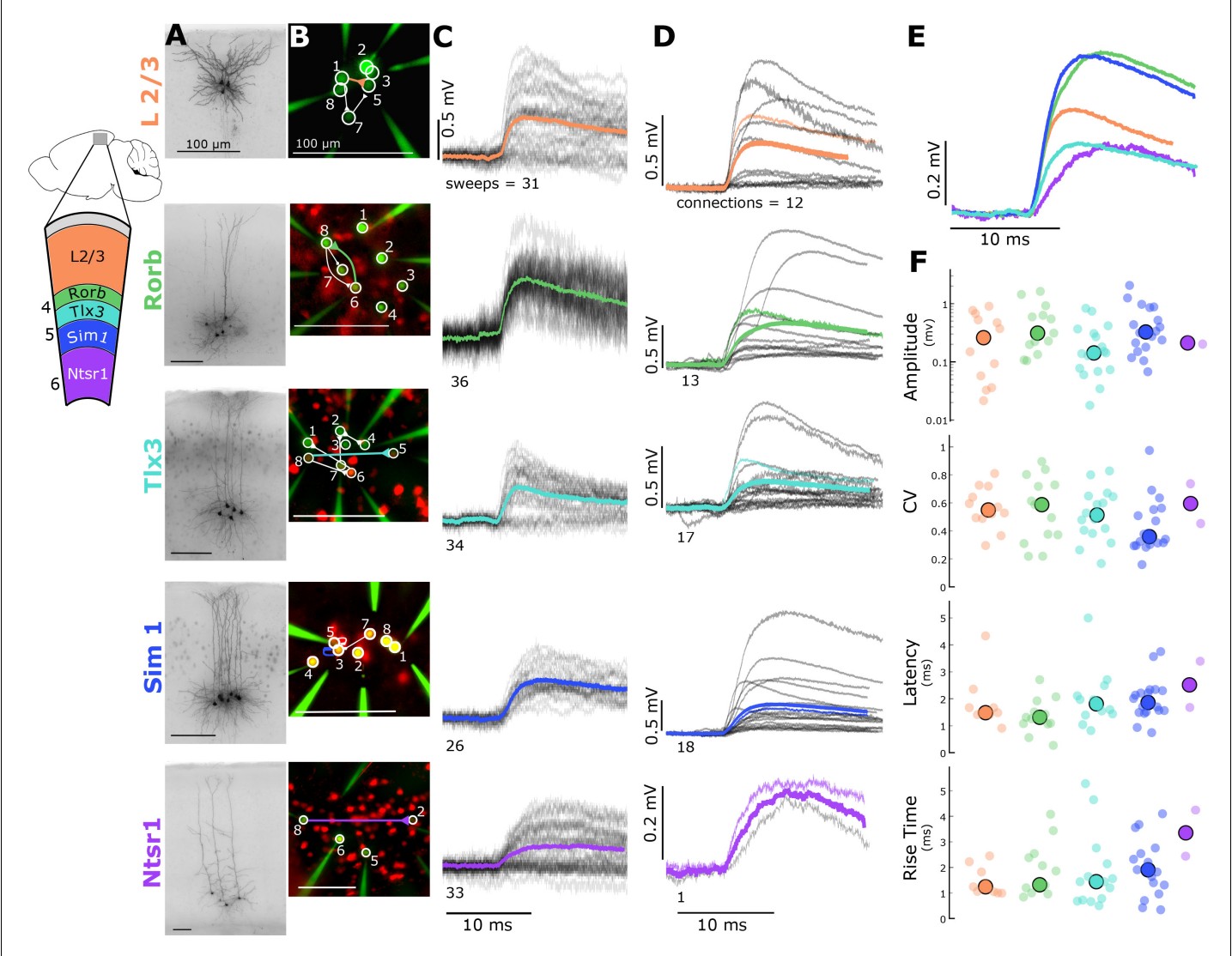

**Figure 1.** Electrophysiological recordings of evoked excitatory synaptic responses between individual cortical pyramidal neurons in mouse primary visual cortex. (A) Cartoon illustrating color, Cre-line, and cortical layer mapping in slice recording region (V1). Example maximum intensity projection images of biocytin-filled pyramidal neurons for L2/3 and each Cre line. (B) Example epifluorescent images of neurons showing Cre-dependent reporter expression and/or dye-filled recording pipettes. Connection map is overlaid on the epifluorescent image (colored: example connection shown in C). (C) Spike time aligned EPSPs induced by the first AP of all ≤ 50 Hz stimulus trains for a single example connection (individual pulse-response trials: grey; average: colored). (D) First pulse average, like in C., for all connections within the synaptic type; grey: individual connections; thin-colored: connection highlighted in C; thick-colored: grand average of all connections. (E) Overlay of grand average for each connection type. (F) EPSP amplitude (in log units), CV of amplitude, latency, and rise time of first-pulse responses for each Cre-type (small circles) with the grand median (large). See *Figure 1— figure supplement 1* for data processing and analysis diagrams.

DOI: https://doi.org/10.7554/eLife.37349.004

The following source data and figure supplement are available for figure 1:

**Source data 1.** Electrophysiological recordings of evoked excitatory synaptic responses between individual cortical pyramidal neurons in mouse primary visual cortex.
DOI: https://doi.org/10.7554/eLife.37349.006

**Figure supplement 1.** Experiment methodology and analysis workflow.
DOI: https://doi.org/10.7554/eLife.37349.005

individual sweeps were included based on a number of criteria, namely a maximum autobias current to reach a holding potential of −70 ± 5 mV, a stable baseline, and absence of spontaneous spiking (see Materials and methods; *Figure 1—figure supplement 1E,F*). A minimum of 5 QC-passed

**Table 2.** Properties of mouse EPSPs.

Median, mean, and standard deviation of EPSP properties plotted in *Figure 1F* for each layer and Cre-type. Number of connections used in the amplitude and CV analysis are found in *Table 1* 'Strength', or for latency and rise time in *Table 1* 'Kinetics'.

| | Amp median (mV) | Amp mean (mV) | Amp SD (mV) | Latency median (ms) | Latency mean (ms) | Latency SD (ms) | Rise Time median (ms) | Rise Time mean (ms) | Rise Time SD (ms) | CV median | CV mean | CV SD |
|---|---|---|---|---|---|---|---|---|---|---|---|---|
| L2/3 | 0.26 | 0.34 | ±0.32 | 1.48 | 1.87 | ±1.0 | 1.24 | 1.45 | ±0.57 | 0.55 | 0.56 | ±0.15 |
| Rorb | 0.31 | 0.54 | ±0.49 | 1.31 | 1.50 | ±0.6 | 1.32 | 1.63 | ±0.94 | 0.59 | 0.55 | ±0.24 |
| Sim1 | 0.33 | 0.52 | ±0.51 | 1.86 | 2.05 | ±0.82 | 1.91 | 1.86 | ±1.1 | 0.36 | 0.43 | ±0.2 |
| Tlx3 | 0.14 | 0.24 | ±0.24 | 1.81 | 2.07 | ±0.74 | 1.44 | 1.35 | ±1.1 | 0.51 | 0.51 | ±0.18 |

DOI: https://doi.org/10.7554/eLife.37349.007

sweeps were required for each connection to be included. *Figure 1C* shows EPSPs recorded from one example connection found in each of the chosen excitatory cell groups. For the large majority of connections, it was not possible to unequivocally distinguish synaptic failures from detection failures, thus we used the mean response from all sweeps (*Figure 1C*) to evaluate the EPSP features.

Consistent with previous reports that recurrent connectivity is weak (*Song et al., 2005*; *Lefort et al., 2009*), we found that a majority of the connections had amplitudes less than 0.5 mV. In this small sample, we did not observe statistical difference in the EPSP amplitudes (*Figure 1E,F*) between groups (KW p=0.07), although there was a trend toward overall smaller Tlx3 EPSP amplitudes (median ± SD 0.14 ± 0.24 mV). The range of amplitudes for layer 2/3 (0.032–0.902 mV), Rorb (0.105–1.626 mV), Sim1 (0.068–1.254 mV), and Tlx3 (0.02–0.833 mV) spanned an order of magnitude. We could not assess the range of recurrent Ntsr1 connections due to the low number of connections measured; however, the amplitude and relatively long latency (*Figure 1F*) are consistent with connections between corticothalamic (CT) layer six neurons in the rat cortex (*West et al., 2006*; *Table 1*). The mean EPSP amplitude was consistently larger than the median (*Table 2*) due to a skewed (long-tailed) distribution of response amplitudes. Similar observations in the rat visual cortex, and mouse somatosensory cortex, has led to the suggestion that rare, large-amplitude connections are important for reliable information processing (*Song et al., 2005*; *Lefort et al., 2009*; *Cossell et al., 2015*). The majority of EPSP latencies were less than 2.5 ms (*Table 2*), and similar across populations (KW p=0.17), consistent with a direct, monosynaptic connection between recorded neurons.

We could not directly quantify synaptic failures and thus calculated the coefficient of variation of synaptic amplitudes (CV; *Figure 1F*) to assess release probability. The CV of each connection describes the variability in a particular response in relation to the mean (ratio of standard deviation to mean) and is negatively correlated with release probability (*Markram, 1997*). The range of coefficient of variation in our data suggests differences in release probability between cell classes and is consistent with STP modeling results (see Figure 6).

## Properties of intralaminar excitatory synaptic signaling in human cortex

To what extent is recurrent connectivity in mouse V1 representative of connectivity in other regions and species? To make this comparison, we performed multipatch recordings from human frontal and temporal cortex. Specimens were collected during surgical resection of epileptic or tumorous tissue, but were distal to the site of pathology. We sampled recurrent intralayer connectivity in all layers containing pyramidal cells. Pyramidal cells were identified by their morphology visualized via biocytin (*Figure 2A*) or fluorescent dye (*Figure 2B*). We found 22 connections between layer 2 pyramidal cells (132 probed), 37 connections between layer 3 pyramidal cells (249 probed), four connections between layer 4 pyramidal cells (123 probed), and 13 connections between layer 5 pyramidal cells (112 probed). We found 1 connection in layer 6 (16 probed connections) but have not yet probed this layer sufficiently to make confident measurements of connection probability or synaptic properties. We selected 1.3 mM $[Ca^{++}]_e$ for our human experiments because of reports that synaptic strength is higher than in mouse and to minimize the complex events that can be initiated by individual spikes in human tissue (*Molnár et al., 2008*) that make identifying monosynaptic connectivity challenging. Indeed we found that human cortex had a higher connectivity rate and mean amplitude

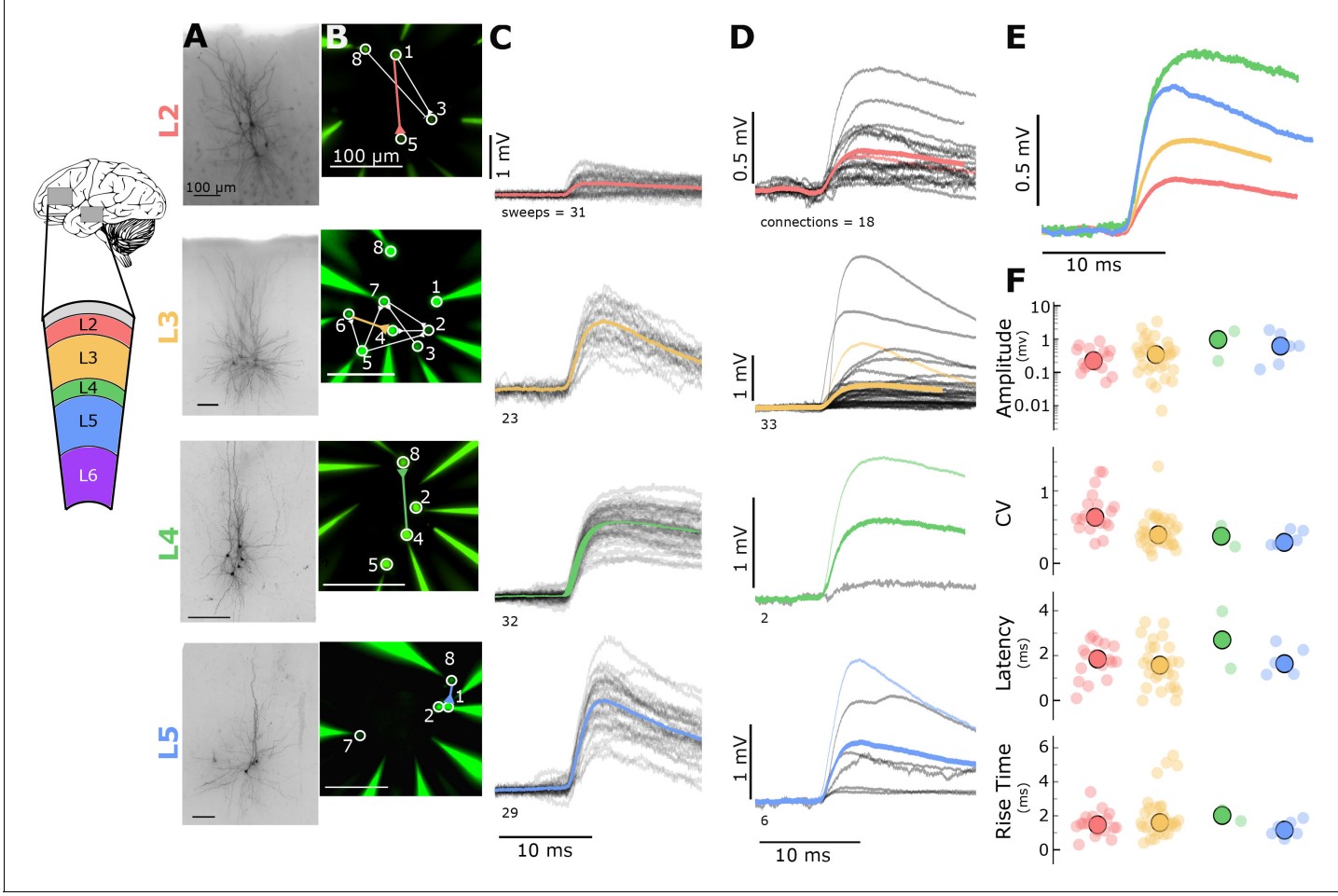

**Figure 2.** Electrophysiological recordings of evoked excitatory synaptic responses between individual human cortical pyramidal neurons. (A) Cartoon illustrating color and cortical layer mapping in slice recording region (temporal or frontal cortex). Example maximum intensity projection images of biocytin-filled pyramidal neurons for layers 2–5. (B) Example epifluorescent images of neurons showing dye-filled neurons and recording pipettes. Connection map is overlaid on the epifluorescent image (colored: example connection shown in C). (C) Spike time aligned EPSPs induced by the first AP of all ≤ 50 Hz stimulus trains for a single example connection (individual pulse-response trials: grey; average: colored). (D) First pulse average, like in C., for all connections within the synaptic type; grey: individual connections; thin-colored: connection highlighted in C; thick-colored: grand average of all connections. (E) Overlay of grand average for each connection type. (F) EPSP amplitude, CV of amplitude, latency, and rise time of first-pulse responses for each layer (small circles) with the grand mean (large circles). See *Figure 1—figure supplement 1* for data processing and analysis diagrams.

DOI: https://doi.org/10.7554/eLife.37349.008

The following source data is available for figure 2:

**Source data 1.** Electrophysiological recordings of evoked excitatory synaptic responses between individual human cortical pyramidal neurons.
DOI: https://doi.org/10.7554/eLife.37349.009

(*Figure 2C,D*) compared to mouse cortex (despite a higher $[Ca^{++}]_e$ in mouse), consistent with previous reports (*Molnár et al., 2008*). Layers 2, 3, and 5 had a sufficient number of connections to characterize strength and kinetics. However, we found no differences in response properties among these three layers (amplitude p=0.22, latency p=0.51, rise time p=0.22, *Table 3*). We did observe differences in CV between layers 2, 3, and 5 (p=0.0004, *Table 3*) suggesting layer-specific differences in release probability of recurrent connections, similar to findings in mouse V1.

It is reasonable to question if the recurrent connectivity we see in tissue from epilepsy and tumor patients differs from that of healthy individuals. Although we cannot rule this out, we saw no significant differences in overall connectivity between tumor and epilepsy-derived specimens (p=0.833, Fisher's Exact Test). We also found recurrent connections within multiple cortical regions and disease

**Table 3.** Properties of human EPSPs.

Median, mean, and standard deviation of EPSP properties plotted in *Figure 1F* for each layer and Cre-line. Number of connections used in the amplitude and CV analysis are found in *Table 1* 'Strength', for latency and rise time in *Table 1* 'Kinetics'.

| | Amp median (mV) | Amp mean (mV) | Amp SD (mV) | Latency median (ms) | Latency mean (ms) | Latency SD (ms) | Rise time median (ms) | Rise time mean (ms) | Rise time SD (ms) | CV median | CV mean | CV sd |
|---|---|---|---|---|---|---|---|---|---|---|---|---|
| L2 | 0.22 | 0.30 | ±0.22 | 1.84 | 1.79 | ±0.78 | 1.47 | 1.53 | ±0.59 | 0.80 | 0.64 | ±0.29 |
| L3 | 0.34 | 0.54 | ±0.68 | 1.57 | 1.58 | ±0.97 | 1.60 | 2.07 | ±1.36 | 0.39 | 0.44 | ±0.23 |
| L4 | 0.97 | 0.97 | ±1.05 | 2.70 | 2.70 | ±1.80 | 2.02 | 2.02 | ±0.47 | 0.37 | 0.37 | ±0.20 |
| L5 | 0.62 | 0.80 | ±0.69 | 1.64 | 1.75 | ±0.59 | 1.16 | 1.23 | ±0.46 | 0.29 | 0.34 | ±0.10 |

DOI: https://doi.org/10.7554/eLife.37349.010

states in the human. Taken together, this may indicate that our results capture a common architecture of the mouse and human microcircuit.

## Detection limit of synaptic responses

When using whole cell recordings to characterize synaptic connectivity, a major limitation is that some EPSPs may be too weak to be detected at the postsynaptic soma. Detection limits are influenced by several factors including the kinetics of EPSPs, the amplitude and kinetics of background noise, the frequency and properties of spontaneous EPSPs, and the number of evoked presynaptic spikes. One consequence is that we expect to generally underestimate connectivity, and in some cases, cell class differences in synaptic strength can be misinterpreted as differences in connectivity. Another consequence is that it may not be possible to obtain an accurate measurement of the distribution of synaptic weights, since the weakest synapses are undetectable.

To address these issues, we characterized the sensitivity of our experiments by testing whether a machine classifier could detect simulated EPSPs of varying, known strength (see Materials and Methods). The classifier was trained to detect connections based on features extracted from the averaged response to evoked spikes (*Figure 3B*; features listed in *Supplementary file 1*) and from the distributions of features measured on individual responses (*Figure 3C*). For each putative connection probed, we collected recordings of background activity when no cells were being stimulated and superimposed EPSP-like deflections. These recordings were then processed for features (Figure 3 – table supplement 1) which were fed to the classifier to generate connectivity predictions. By testing several sets of artificial EPSPs in which we systematically varied the average amplitude, we were able to measure the relationship between EPSP strength and the probability that a connection could escape detection (*Figure 3D*).

This analysis provides, for every connection that we probed, an estimate of the minimum detectable EPSP amplitude (*Figure 3A*). Recordings with low background noise and adequate averaging will generally allow the detection of very small EPSPs (*Figure 3D*, top panel has a detection limit of 10–20 µV), whereas lower quality recordings will have higher detection thresholds and will report lower connectivity rates (*Figure 3D*, bottom panel has a poor detection limit near 100 µV). Likewise, EPSPs with shorter rise time (or other properties that distinguish the EPSP from background) are more likely to be detected (*Figure 3D*). These results confirm that the differences in experimental protocol between studies (for example, the number of presynaptic spikes evoked for each connection) can have a substantial impact on the apparent connectivity reported, but also suggests that future studies could reconcile these differences by carefully characterizing their detection limits.

The results of this analysis also suggest a means of estimating the shape of the underlying distribution of synapse strengths at the low end, where synapses become more difficult to detect. *Figure 3E* shows the distribution of EPSP amplitudes across all detected synapses (light grey area) as well as the curve representing the probability that synapses would be detected as a function of EPSP amplitude (red line). Dividing the measured distribution by the probability of detection yields a corrected distribution (dark grey) with an overall 10% increase in connectivity. Although this estimate becomes unstable as the detection probability nears zero, an interesting result is that the left edge of the distribution trends downward even in the region where detection probability is high, suggesting that the sensitivity of our experiments is adequate to capture the majority of synapses and that

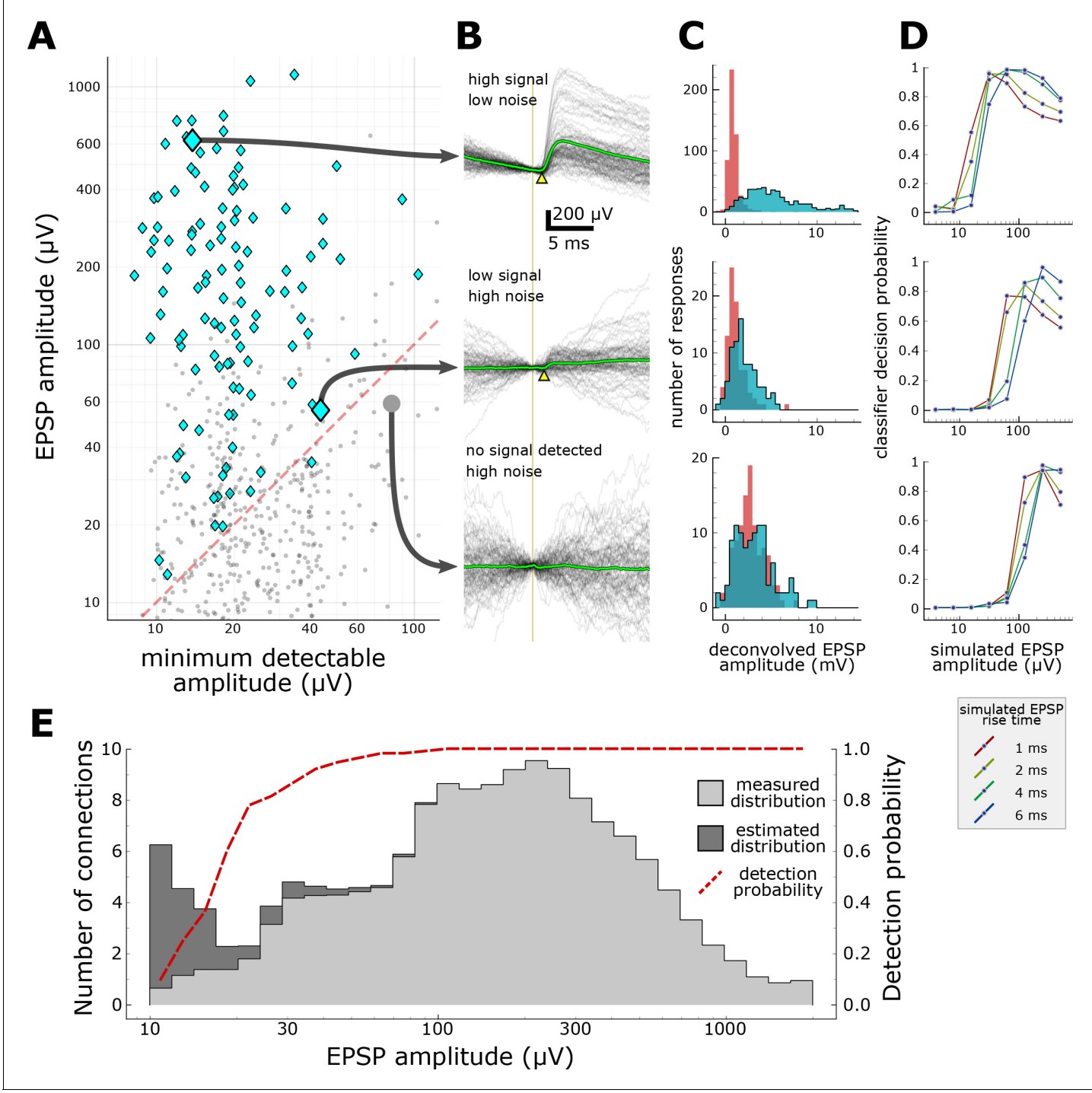

**Figure 3.** Characterization of synapse detection limits. (**A**) Scatter plot showing measured EPSP amplitude versus minimum detectable amplitude for each tested pair. Detected synapses (manually annotated) are shown as blue diamonds; pairs with no detected EPSPs are grey dots. The region under the red dashed line denotes the region in which synaptic connections are likely to be misclassified as unconnected. Three example putative connections are highlighted in A and described further in panels B-D. One connection (top row) was selected for its large amplitude PSP and low background noise. Another connection (middle row) is harder to detect (PSP onset marked by yellow arrowhead) due to low amplitude and high background noise. The bottom row shows a recorded pair that was classified as unconnected. (**B**) A selection of postsynaptic current clamp recordings in response to presynaptic spikes. Each row contains recordings from a single tested pair. The vertical line indicates the time of presynaptic spikes, measured as the point of maximum dV/dt in the presynaptic recording. Yellow triangles indicate the onset of the EPSP. (**C**) Histograms showing distributions of peak response values measured from deconvolved traces (see Materials and methods). Red area indicates measurements made on background noise; blue area indicates measurements made immediately following a presynaptic spike. (**D**) Characterization of detection limits for each

*Figure 3 continued on next page*

 Research article

*Figure 3 continued*

example. Plots show the probability that simulated EPSPs would be detected by a classifier, as a function of the rise time and mean amplitude of the EPSPs. Each example has a different characteristic detection limit that depends on the recording background noise and the length of the experiment. (E) An estimate of the total number of false negatives across the entire dataset. The measured distribution of EPSP amplitudes is shown in light grey (smoothed with a Gaussian filter with σ = 1 bin). The estimated correction show in dark grey is derived by dividing the measured distribution by the overall probability of detecting a synapse (red dashed line) at each amplitude. See **Supplementary file 1** for features included in classifier.
DOI: https://doi.org/10.7554/eLife.37349.011

The following source data is available for figure 3:

**Source data 1.** Characterization of synapse detection limits.
DOI: https://doi.org/10.7554/eLife.37349.012

we have accurately represented the underlying distribution of synaptic strengths. We are cautious, however, in our interpretation of this result–the analysis relies on several assumptions about the behavior of the classifier and the realism of the simulated EPSPs. Ultimately, the approach must be validated against a larger dataset.

## Connection probability of excitatory synapses

Estimates of connectivity vary widely across studies, in part due to methodological differences. In addition to the effects of detection sensitivity described above, estimated connection probability is affected by the intersomatic distances over which connections are sampled. This spatial distribution of connections may also offer insight into the organization of functional microcircuits. In mouse, connectivity in layer 2/3, Rorb, and Sim1 neurons within 100 μm was similar (~10%; **Figure 4**, left; connected/probed, L2/3: 13/130, Rorb: 18/247, Sim1: 41/527). However, within this range, Tlx3 connectivity was markedly lower (~5%; Tlx3: 36/746). Consistent with previous experiments in rat L6 CT neurons (**West et al., 2006**), Ntsr1 connectivity was very sparse as only two connections were detected (out of 313 probed) and were relatively far apart (intersomatic distance of 163 and 127 μm; Bonferroni corrected p<0.01 relative to all other groups). Most connectivity versus distance profiles (**Figure 4B**) showed a progressive reduction in the connection probability with increasing distance. We did not carry out an analysis of reciprocal (bi-directional) connectivity because we lacked the statistical power to detect differences between our classes. Furthermore, measuring reciprocal connectivity at the cell class level can yield misleading results (**Hoffmann and Triesch, 2017**).

In human cortex, layers 2 and 3 had a similar connection probability (~15%; layer 2: 22/132; layer 3: 37/249; **Figure 4C**), while layer 4 recurrent connectivity was much smaller (3.3%, 4/123, Bonferroni corrected p<0.02 relative to all other layers). More data are needed to accurately resolve the distance dependence of recurrent connectivity in the human (**Figure 4D**).

Utilizing a multipatch technique limits our ability to probe connectivity at high density and long distances. We used two-photon optogenetic stimulation to overcome these limitations, which allows for focal stimulation of many (mean = 56 cells, range = 8–117 cells) presynaptic cells in a single experiment, and critically, allows probing distances greater than is generally feasible with multipatch experiments. ReaChR expressing Tlx3-Cre neurons in layer 5 were photo-stimulated with a two-photon laser while one or two putative postsynaptic cells were monitored in whole-cell current clamp configuration (**Figure 4—figure supplement 1**; **Figure 4—figure supplement 1A**).; With this technique, using 12 mice, we found a similar connection probability over the distance range of multipatch experiments (4/136, 2.15%) and reduced connectivity at extended distances up to 785 μm (**Figure 4E**; 13/1594, 0.82%) with the furthest connection found at 459 μm.

## Short-term plasticity of excitatory synapses

For a subset of synaptic connections in mouse cortex (**Figure 1—figure supplement 1G**, **Table 1** STP), we characterized and modeled short-term synaptic dynamics. We probed short-term dynamics with stimulus trains consisting of 8 pulses to induce STP, followed by a variable delay and 4 more pulses to measure recovery (**Figure 5A**, left). The 8 initial pulses allowed responses to reach a steady state, from which we could characterize the extent of depression (or facilitation) at frequencies from 10 to 100 Hz. The 50 Hz stimulation protocol had additional recovery intervals ranging from 250 to 4000 ms (**Figure 5A**, right). Although typical experiments use a single delayed pulse to measure

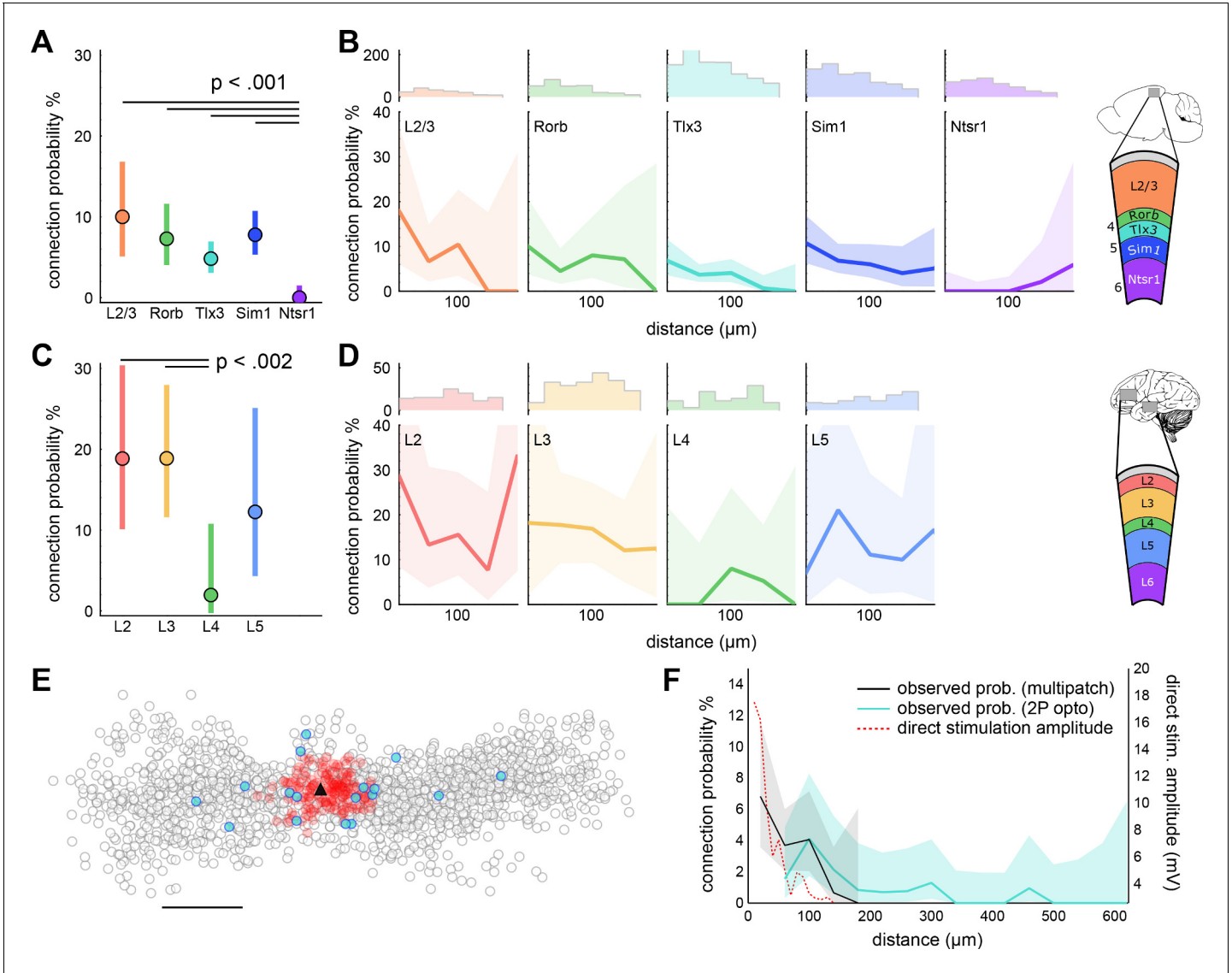

**Figure 4.** Distance dependent connectivity profiles of mouse and human E-E connections. (A) Recurrent connection probability and distribution of connections for mouse -linesand layer 2/3. Mean connection probability (filled circles) and 95% confidence intervals (bars) for connections probed within 100 μm (*n* connections in **Table 1**). (B) Connection probability over distance for mouse Cre-lines and layer 2/3. *Top*: Histogram of putative connections probed. *Bottom*: Mean connection probability (thick line) with 95% confidence intervals (shading) binned in 40 μm increments. (C) Like-to-like connection probability and distribution of connections between human pyramidal neurons. Mean connection probability (filled circles) and 95% confidence intervals (bars) for connections probed within 100 μm. (D) Connection probability over distance for human pyramidal neurons, formatted as in panel B. (E) Tlx3-Tlx3 connection probability measured by two-photon mapping. X-Y distance distribution of connections probed onto a postsynaptic cell (black triangle), detected presynaptic neurons (filled circles), no connection detected (empty circles), and direct event artifact due to undesired activation of opsin in the dendritic arbor of the recorded cell (red circles). (F) Connection probability and stimulation artifact over distance measured by two-photon mapping. Mean connection probability vs. distance (blue line; starting at 50 μm) with 95% confidence intervals (shading) and direct event artifact amplitude vs. distance (dotted red line) for Tlx3-Tlx3 connections probed with two-photon stimulation. See **Figure 4—figure supplement 3** for distribution of connectivity as a function of cortical slice position and cell depth. See **Figure 4—figure supplements 1,2** for details on two-photon connectivity experiments.

DOI: https://doi.org/10.7554/eLife.37349.013

The following source data and figure supplements are available for figure 4:

**Source data 1.** Distance dependent connectivity profiles of mouse and human E-E connections.

DOI: https://doi.org/10.7554/eLife.37349.017

**Figure supplement 1.** Intralaminar connectivity rates were unaffected by recording depth and medial-lateral position in V1.

DOI: https://doi.org/10.7554/eLife.37349.014

*Figure 4 continued*

**Figure supplement 1—source data 1.** Intralaminar connectivity rates were unaffected by recording depth and medial-lateral position in V1.

DOI: https://doi.org/10.7554/eLife.37349.018

**Figure supplement 2.** Characterization of two-photon photostimulation.

DOI: https://doi.org/10.7554/eLife.37349.015

**Figure supplement 2—source data 1.** Characterization of two-photon photostimulation.

DOI: https://doi.org/10.7554/eLife.37349.019

**Figure supplement 3.** Two-photon optogenetic mapping details.

DOI: https://doi.org/10.7554/eLife.37349.016

**Figure supplement 3—source data 1.** Two-photon optogenetic mapping details.

DOI: https://doi.org/10.7554/eLife.37349.020

recovery, we used a short train of 4 pulses to improve sensitivity in modeling the recovered state of the synapse. *Figure 5B* shows average synaptic responses to a 50 Hz stimulus with eight initial pulses followed by four pulses at a 250 ms delay from individual Sim1-Sim1 connections shown in grey, with the grand average overlaid (blue). We used exponential deconvolution (*Figure 5B*, middle; *Equation 2*) to estimate the amplitudes of individual PSPs in the absence of temporal summation (arising from the relatively long cell membrane time constant).

We measured the peak amplitude of the deconvolved response for every pulse (blue dots) and normalized to the first pulse in the train in order to characterize short-term dynamics across four frequencies. *Figure 5C* (top) highlights frequency dependent depression in recurrent Sim1 connections. Across cell classes we saw depression in Rorb, Tlx3, and Sim1 synapses, whereas layer 2/3 synapses showed modest facilitation on average (*Figure 5C*, bottom left). The two Ntsr1 connections (data not shown) also showed facilitation as has been previously reported for layer 6 CT neurons (*West et al., 2006*). The amplitude ratio of the last pre-recovery pulse (8) to the first, for individual connections (*Figure 5D*, left), highlights the heterogeneity in layer 2/3 dynamics where some synapses depressed strongly (ratio < 1) and others facilitated (ratio > 1). Individual dynamics in Rorb, Tlx3, and Sim1 generally showed more depression than layer 2/3 (KW p=0.04). In addition to probing induction of short-term plasticity, we also measured recovery from short-term effects at various time delays (*Figure 5B*, bottom, blue dots) for layer 2/3, Rorb, Sim1, and Tlx3 connections (*Figure 5C*, bottom). All types showed a similar time-course of recovery as measured by the ratio of the first recovery pulse (9) to the first induction pulse (*Figure 5D*, right).

Although we observed depression in both layer 5 classes, previous reports in rat somatosensory cortex showed that recurrent connections between thick tufted layer 5b cortical neurons can facilitate (*Reyes and Sakmann, 1999*; *Lefort and Petersen, 2017*). The availability of free calcium in the presynaptic terminal impacts release probability and thus short-term dynamics of a synapse (*Rozov et al., 2001*). We hypothesized that 0.3 mM EGTA in the intracellular solution may buffer presynaptic calcium accumulation and reduce the magnitude of facilitation observed in our layer 5 connections. We tested this in a subset of recurrent Sim1 connections (23 connections found/210 probed) and Tlx3 connections (five connections found/52 probed) in which there was no EGTA in the internal solution ($[Ca^{++}]_e$ = 2.0mM). Over the course of a 50 Hz train, we saw overall depression in both Tlx3 and Sim1 (*Figure 5—figure supplement 1A*) such that the 8:1 pulse ratio was statistically indistinguishable in the two EGTA concentrations (Tlx3 KS test p=0.47; Sim1 KS test p=0.35; *Figure 5—figure supplement 1B*; *Table 4*). We did observe, however, that Tlx3 synapses show transient facilitation during the second pulse of the train (paired-pulse ratio; KS test p=0.04; *Figure 5—figure supplement 1C*, top; *Table 4*). We did not see a significant difference in paired-pulse ratio for Sim1 connections in the absence of EGTA (KS test p=0.35; *Figure 5—figure supplement 1C*, bottom; *Table 4*). This suggests that Tlx3 and Sim1 cell classes have different synaptic dynamics but are nonetheless dominated by depression.

To more fully capture the dynamic processes contributing to short-term plasticity and how they may differ among connection types we turned to a model of short-term dynamics which has been well described (*Hennig, 2013*; *Mongillo et al., 2008*; *Richardson et al., 2005*). For homogeneous populations, we expect that applying the model to an average response will accurately represent the short-term dynamics of a particular cell class. Layer 2/3, Rorb, Tlx3, and Sim1 synapses were modeled with depression (*Equation 4*, Materials and methods) and use-dependent replenishment

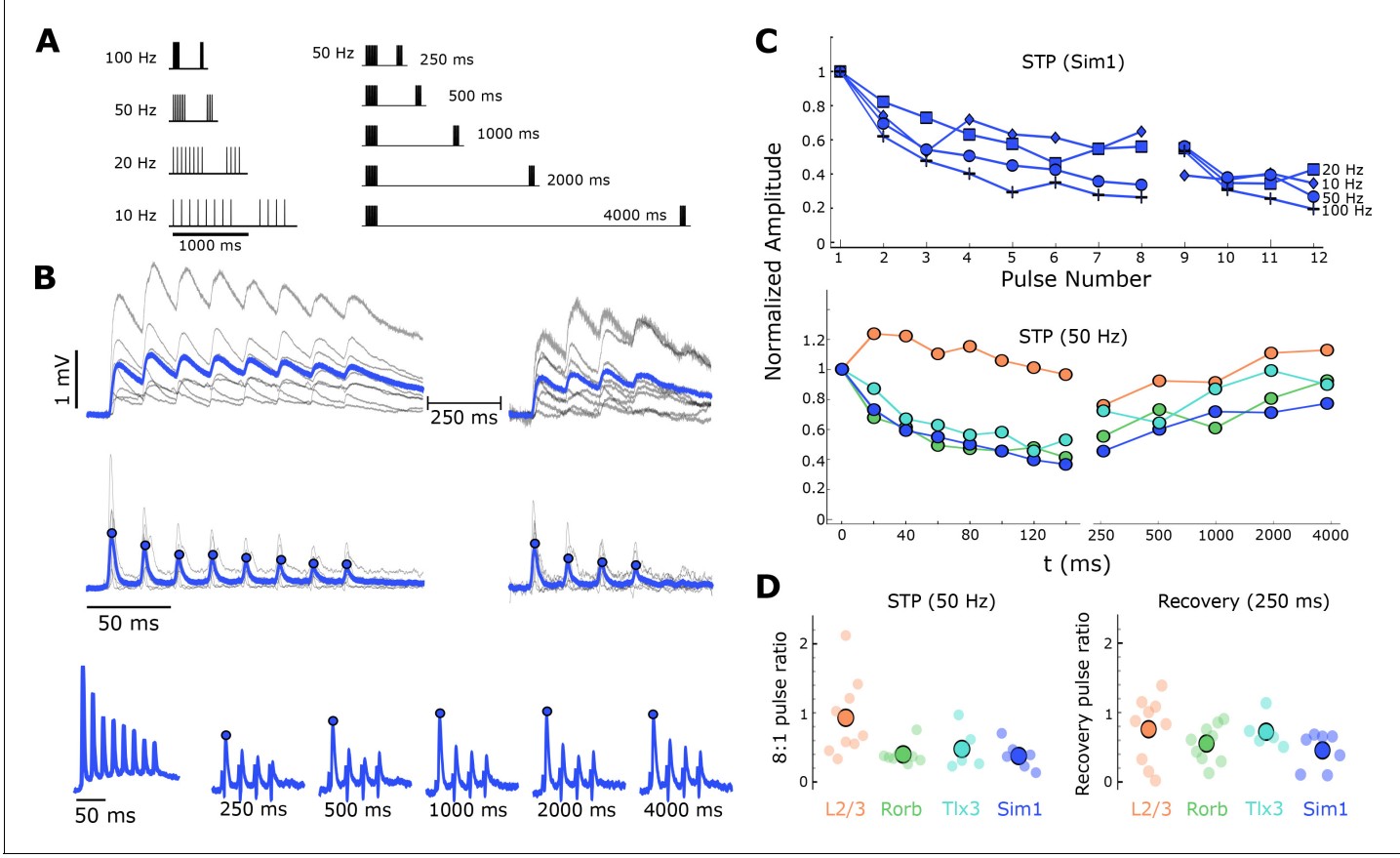

**Figure 5.** Short-term dynamics of mouse recurrent connections by Cre-line and layer (*n* in *Table 1* 'STP'). (**A**) Schematic of STP and STP recovery stimuli. (**B**) Sim1-Cre EPSPs in response to a 50 Hz stimulus train (top; eight induction pulses and four recovery pulses delayed 250 ms; individual connection: gray traces; blue: Sim1-Cre average EPSP at 50 Hz). Exponential deconvolution followed by lowpass filter of EPSPs above (middle, filled circles: pulse amplitudes in C). Exponential deconvolution of 50 Hz stimulus with all five recovery time points in A (bottom, filled circles: pulse amplitudes in C). (**C**) The mean normalized amplitude of deconvolved response versus pulse number at multiple stimulation frequencies for Sim1-Cre (top). Normalized amplitude of the deconvolved response at 50 Hz with first recovery pulse at each interval for each Cre-line and L2/3 connections (bottom). (**D**) The depth of depression during 50 Hz induction (left) as measured by the amplitude ratio of the 8th to 1st pulse for each Cre-line and layer (small circles) and grand mean (large circles). Amount of recovery at 250 ms latency (right) for each Cre-line and layer (small circles) and grand mean (large circles). See *Figure 5—figure supplement 1* for results of STP at different EGTA concentrations and *Figure 1—figure supplement 1* for data analysis diagram.

DOI: https://doi.org/10.7554/eLife.37349.021

The following source data and figure supplements are available for figure 5:

**Source data 1.** Influence of internal EGTA on short-term dynamics.
DOI: https://doi.org/10.7554/eLife.37349.023
**Figure supplement 1.** Influence of internal EGTA on short-term dynamics.
DOI: https://doi.org/10.7554/eLife.37349.022
**Figure supplement 1—source data 1.** Influence of internal EGTA on short-term dynamics.
DOI: https://doi.org/10.7554/eLife.37349.024

(*Equation 5*, Materials and methods). For the more homogeneous cell classes (Rorb, Tlx3, and Sim1), the model performed well (*Figure 6A*, Sim1 connections $r^2$ = 0.845, *Table 5*) in capturing depression during the eight initial pulses at various frequencies (*Figure 6A*) as well as modeling recovery at various delays for 50 Hz stimuli (*Figure 6A*, open circles). Heterogeneity among layer 2/3 synapses made it difficult to constrain the model and thus were not included in further analysis. From this model, we can extract free parameters such as $P_0$ which estimates release probability and $\tau_{r0}$ which estimates the time course of recovery from depression. Rorb connections had the largest release probability ($P_0$ = 0.30, *Figure 6B*, left) which is consistent with faster entry into depression

**Table 4.** Mean and standard deviation of 8:1 ratio at 50 Hz and 9:1 ratio at 50 Hz and 250 ms delay for individual synapses. Unless noted, EGTA was 0.3 mM and n's are listed in *Table 1*, STP.

| | 8:1 pulse ratio (50 Hz) mean ± SD | Recovery (9:1) ratio (250 ms) mean ± SD | Paired-pulse ratio (50 Hz) mean ± SD |
|---|---|---|---|
| L2/3 | 0.92 ± 0.57 | 0.76 ± 0.48 | 1.14 ± 0.63 |
| Rorb | 0.39 ± 0.14 | 0.55 ± 0.26 | 0.66 ± 0.13 |
| Sim1 | 0.37 ± 0.18 | 0.46 ± 0.26 | 0.73 ± 0.17 |
| Tlx3 | 0.48 ± 0.32 | 0.72 ± 0.24 | 0.8 ± 0.15 |
| Sim1 (0 EGTA), n = 6 | 0.27 ± 0.17 | N/A | 0.68 ± 0.22 |
| Tlx3 (0 EGTA), n = 2 | 0.51 ± 0.12 | N/A | 1.15 ± 0.03 |

DOI: https://doi.org/10.7554/eLife.37349.025

(*Figure 5C*, bottom pulse 2). Conversely, Tlx3 connections had a lower initial release probability ($P_0$ = 0.16) and recovered more quickly ($\tau_{r0}$ = 0.736 s). *Table 5* shows full results of the model across these three types as well as calculated paired Z-scores (*Equation 6*) for $P_0$ and $\tau_{r0}$.

## Discussion

We leveraged the sub-millisecond sampling, high gain, and low noise of multipatch recordings to investigate the functional connectivity and short-term dynamics of recurrent synapses in the adult mouse and human cortex. We observed sparse recurrent connections between excitatory neurons in layers 2/3 through 6 in adult mouse visual cortex and layers 2 through 6 of adult human cortex. We supplemented mouse multipatch experiments with high-throughput 2P optogenetic stimulation to sample connectivity at greater distances than is generally feasible using the multipatch approach. Most excitatory recurrent connections in mouse cortex were dominated by short-term synaptic depression.

Estimates of connectivity derived from multipatch experiments in brain slices should be considered as a lower bound on the underlying population connectivity due to sensitivity to false negatives from several sources. These effects may contribute to differences in reported connectivity across studies. A fraction of synaptic connections are expected to be severed during slicing; one estimate of connectivity perturbed by slicing approaches 50% (*Levy and Reyes, 2012*). The effect on measured connection probability depends on the thickness of the slice, the depth of recorded cells from the cut surface, the morphology of recorded cells, and the distance between them. Although we minimize lost connections by patching deep in the slice (>40 μm; *Figure 4—figure supplement 2C, D*) and by selecting cells in close proximity, this is still a likely source of false negatives in our data. Another fraction of synapses are expected to be either too weak (*Isaac et al., 1995*) or too distal from the recording pipette to be detected. The magnitude of this effect is difficult to estimate, but our initial analysis hints that our methods are sensitive enough to capture the majority of synapses. To obtain more accurate estimates of connectivity, it will be necessary to combine these results with other methods such as *in vivo* multipatch recordings, transsynaptic tracing, and serial section electron microscopy. These methods are also limited, but in each case the constraints are different and potentially complementary.

There is a wide range of reported rates of recurrent connectivity among excitatory neurons in rodent studies (*Thomson et al., 2002*; *Hofer et al., 2011*; *Jiang et al., 2015*). One suggestion is that differences between the juvenile and adult rodent can explain the variance (*Jiang et al., 2016*). To avoid changes associated with development, we carried out our experiments in the adult (about two months old) cortex. Nevertheless, our conclusion that recurrent connectivity is sparse but not absent is similar to results from experiments in other adult (*Reyes and Sakmann, 1999*; *Lee et al., 2016*) and juvenile animals (*Mason et al., 1991*; *Holmgren et al., 2003*; *Song et al., 2005*; *Sjöström et al., 2001*; *Morishima et al., 2011*; *Perin et al., 2011*; *Lefort et al., 2009*; *Levy and Reyes, 2012*; *Cossell et al., 2015*). A notable difference, however, is that we never observed rates of recurrent connectivity as high or synaptic amplitudes as large as those reported in juvenile rodents, consistent with the observation that the rate of recurrent connectivity and synaptic strength declines with age (*Bourgeois and Rakic, 1993*; *Reyes and Sakmann, 1999*).

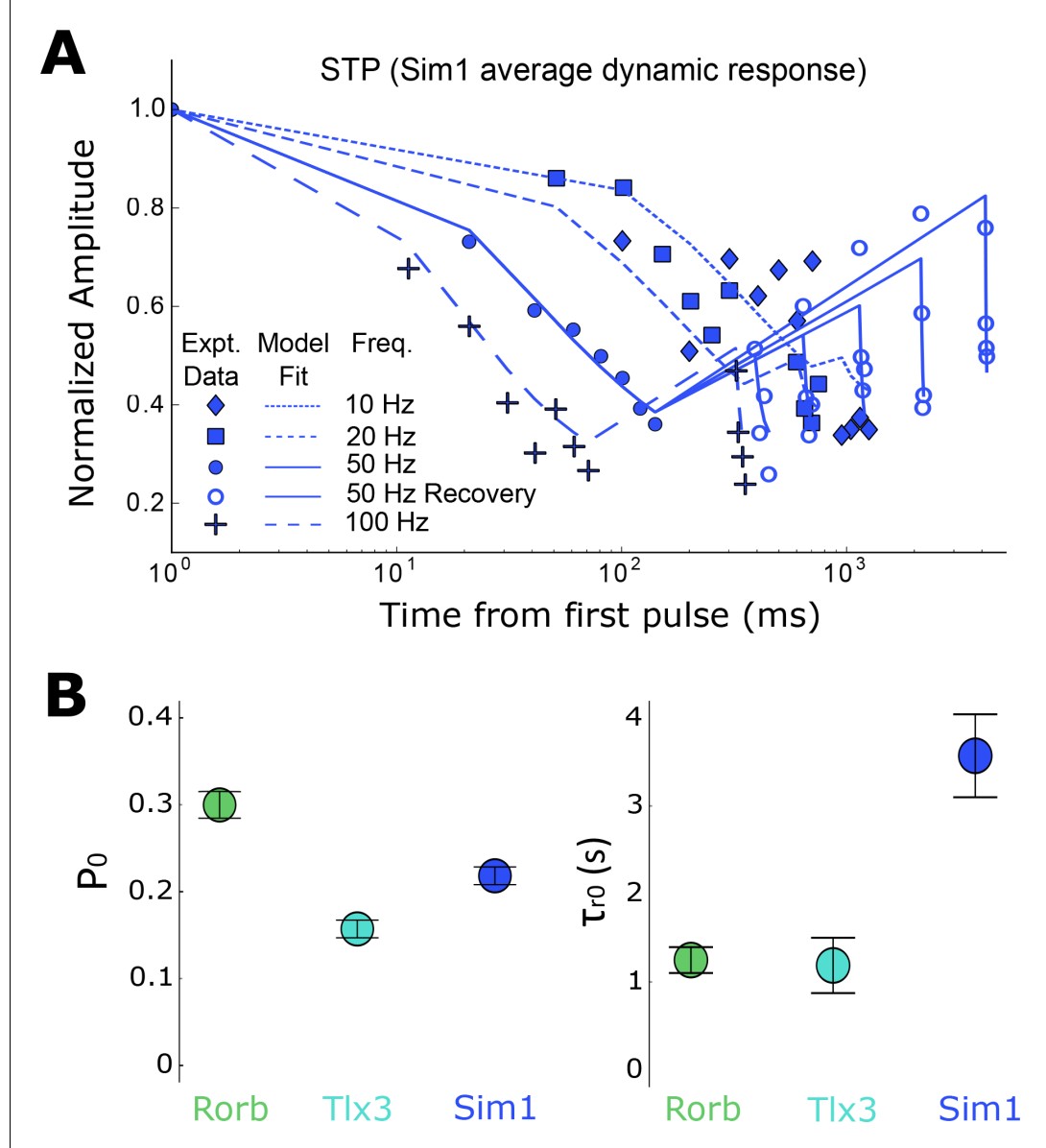

**Figure 6.** Modeling of short-term depression in recurrent Rorb, Sim1, and Tlx3 connections (*n* in *Table 1* 'STP'). (**A**) Sim1 average dynamic response; Same data as in *Figure 5C*, top plotted on a log-X time scale with modeling fits overlaid. (**B**) Results of model for parameters $P_0$ and $\tau r_0$. Values are means with standard error of the covariance matrix. Paired Z-scores (*Equation 6*) in *Table 5*.

DOI: https://doi.org/10.7554/eLife.37349.026

All excitatory cell classes investigated in this study, in human and in mouse, exhibited recurrent connectivity but the rate of recurrent connectivity depended on the class. For example, in two sub-cortically projecting neuron classes, the connectivity rate was approximately 10% in Sim1 (layer 5) expressing cells, whereas Ntsr1 expressing cells (layer 6) had the lowest rate of intralaminar connectivity among all excitatory cell classes tested in this study. This low connectivity is consistent with previous work on layer 6 CT cells using multipatch (*West et al., 2006*) and single cell rabies tracing (*Vélez-Fort et al., 2014*). Our results suggest that recurrent connectivity may be a general property of excitatory neurons that is regulated depending on the target region.

How do the rates of recurrent connectivity in mouse compare to other species? In human cortex, we and others (*Molnár et al., 2008*) find that the frequency of connectivity among excitatory neurons is at least two and half times greater than the highest connectivity rate observed in the adult

**Table 5.** Model parameter values and statistics for Rorb, Sim1, and Tlx3 recurrent connections.

Parameter values are from the model performed on the grand mean for each connection type with the standard error of the covariance matrix. The Z-score was computed following *Equation 6*; the Z-score between each possible pair should be read as a matrix with the corresponding Cre-line in the row. Number of connections used in this analysis in *Table 1* 'STP'.

| Connection type/Model Parameter | $\tau_{r0}$ (sec ± SE) | $P_0$ (±SE) | $\tau_{FDR}$ (ms ± SE) | $\alpha_{FDR}$ (±SE) | $r^2$ | Rorb Z-score | Sim1 Z-score | Tlx3 Z-score |
|---|---|---|---|---|---|---|---|---|
| Rorb | 1.26 ± 0.29 | 0.30 ± 0.03 | 130.6 ± 56.8 | 0.85 ± 0.09 | 0.836 | N/A | $P_0 = 2.02$ | $P_0 = 3.55$ |
| Sim1 | 3.55 ± 0.93 | 0.22 ± 0.02 | 269.4 ± 128.2 | 0.77 ± 0.12 | 0.836 | $\tau_{r0} = 2.31$ | N/A | $P_0 = 2.12$ |
| Tlx3 | 1.20 ± 0.62 | 0.16 ± 0.02 | 276.3 ± 213.2 | 0.47 ± 0.09 | 0.737 | $\tau_{r0} = 1.12$ | $\tau_{r0} = 2.79$ | N/A |

DOI: https://doi.org/10.7554/eLife.37349.027

mouse. It may be that high rates of recurrent connectivity in human are a circuit feature of higher mammals as this is also reported to be the case in cat and monkey (*Kisvárday et al., 1986*; *McGuire et al., 1991*; *Bopp et al., 2014*). Another possibility, however, is that the observed difference between mouse and human is more related to the different cortical regions. Future work will examine how relatively high recurrent connectivity might be counterbalanced by short-term dynamics and/or inhibitory feedback.

In our study, the distance-dependent connection probability profiles fall off with distance and are consistent with connectivity depending on the extent of overlap between neighboring axons and dendrites (*Peters, 1979*; *Binzegger et al., 2004*, Braitenberg and Schüz 1998, *van Pelt et al., 2013*). It is unlikely that the distance-dependent connectivity profiles we observed are an artifact of tissue preparation as it has previously been demonstrated that the truncation of neuronal processes reduces overall connectivity but maintains the spatial pattern of connections (*Stepanyants et al., 2009*; *Levy and Reyes, 2012*). Further effort is required to determine whether our connectivity profiles cannot be predicted from neuronal morphology as has been observed in, for example, intralaminar connections between corticostriatal neurons (*Brown and Hestrin, 2009*).

Like most similar studies (e.g. *Mason et al., 1991*; *Markram, 1997*; *Sjöström et al., 2001*) that have preceded this work, we have measured connectivity and synaptic properties within broad cell classes that include multiple subtypes. Although we found differences between these classes, we expect that in some cases, important cell-type-dependent differences will be obscured by this approach. Initially, we can mitigate this by characterizing the distribution of properties and relating these to morphological and electrophysiological properties that relate to narrower cell classes or types. However, transcriptomic analysis (*Tasic et al., 2016*; *Chevée et al., 2018*) has refined and expanded the catalog of cell types, necessitating the development of new methods to measure synaptic properties with greater cell type resolution and with enough throughput to address the sizeable combinatorial space of connections between types.

In the primary visual cortex of the adult mouse, we found that recurrent connections, except for Ntsr1 connections and a subset of L2/3 connections, depressed after eight action potentials across a range of frequencies. Although depression dominates the net response, it is important to consider that facilitation, depression, and other mechanisms may simultaneously contribute to overall short-term dynamics. Furthermore, the availability of calcium in the presynaptic terminal can strongly influence the balance between mechanisms in a cell-class-dependent manner (e.g., Tlx3 vs. Sim1; *Figure 5—figure supplement 1*). Despite differences in intracellular EGTA buffering between our study and others, the STP we observed in mouse recurrent synapses is mostly consistent with previous work (*West et al., 2006*; *Lefort and Petersen, 2017*) with the primary exception being the lack of facilitation in paired pulses in layer 5 neurons. Interestingly, when facilitation was unmasked by lowering the intracellular EGTA concentration it was not in the mouse line with thick-tufted pyramidal neurons (Sim1; *Gouwens and Berg, 2018*) as we expected from work in other brain regions (*Reyes and Sakmann, 1999*; *Lefort and Petersen, 2017*), suggesting that the recurrent connectivity STP may differ across brain regions for the same projection class.

Ultimately, we seek a description of the cortical circuit from which mechanistic computational models can be built and hypotheses about cortical function can be tested. Although many parts of the circuit have been described in the past, incompatibilities between experiments have made it difficult to assemble a complete, coherent picture of the whole. We have taken steps toward ensuring

that our results can be interpreted in the context of future experiments, but more work is needed to generate a consistent description of the cortical circuit. To that end, we have begun a large-scale project to replicate these measurements across a wider variety of cell types in the mouse and human cortex; the results of our early-stage data collection presented here suggest that systematic and standardized characterization will provide a detailed, quantitative, and comprehensive description of the circuit wiring diagrams and will facilitate the investigation of circuit computation.

## Materials and methods

### Animals and tissue preparation

Adult mice of either sex (mean age P46.7 ± 6.4; SD) were housed and sacrificed according to protocols approved by the Institutional Animal Care and Use Committee at the Allen Institute (Seattle, WA), in accordance with the National Institutes of Health guidelines. Transgenic mouse lines were used for experimentation and chosen based on cortical layer specific expression and/or known projection patterns. In the following mouse lines, subpopulations of excitatory neurons are selectively labeled with fluorescent reporters (tdTomato or GFP): Tlx3-Cre_PL56;Ai14 (n = 57, mean age ± SD P45.8±5.7), Sim1-Cre_KJ18;Ai14 (n = 20, P50.8 ± 8.2), Rorb-T2A-tTA2;Ai63 (n = 28, P45.8 ± 3.4), Ntsr1-Cre_GN220;Ai140 (n = 13, P43.5 ± 2.8) (Allen Institute; see also http://connectivity.brain-map.org/transgenic). Two drivers, Sim1-Cre (subcortical projecting; CS or PT type; Allen Brain Atlas, http://connectivity.brain-map.org/) and Tlx3-Cre, (corticocortical projecting; CC or IT type; *Kim et al., 2015*), were used to label layer 5 pyramidal cells, in order to sample projection-specific subpopulations. We did not utilize transgenic labeling for targeting layer 2/3 pyramidal cells (n = 11, P47.8 ± 8), but instead relied on *post-hoc* morphological analysis. For optogenetic experiments, Tlx3-Cre driver mice were bred with ROSA26-ZtTA/J mice (Jackson Laboratory) and Ai136 mice (*Daigle et al., 2018*), in which a fusion of the ReaChR opsin (*Lin et al., 2013*) with EYFP is expressed from the TIGRE locus (*Zeng et al., 2008*) in a Cre- and tTA-dependent manner. Previous studies have emphasized the differences in cortical connectivity particularly at older ages. To assess whether age impacted the results reported here, a subset of experiments were repeated for Sim1 connections in older mice (mean age 61 ± 1; SD, n = 10). We saw no difference in recurrent connectivity rate (<100 μm; P40-60: 36/423,>P60: 15/269, Fisher's p=0.23) or response amplitude (P40-60: 0.53 ± 0.12 mV,>P60: 0.59 ± 0.2 mV, p=0.98 KS test) among across the two time points.

To facilitate comparisons across pipeline datasets and for logistical considerations, we adopted the slicing methods established for cell-types pipelines (*Gouwens and Berg, 2018*); http://celltypes.brain-map.org/). Animals were deeply anesthetized with isoflurane and then transcardially perfused with ice-cold oxygenated artificial cerebrospinal fluid (aCSF) containing (in mM): 98 HCl, 96 N-methyl-d-glucamine (NMDG), 2.5 KCl, 25 D-Glucose, 25 NaHCO$_3$, 17.5 4-(2-hydroxyethyl)−1-piperazineethanesulfonic acid (HEPES), 12 N-acetylcysteine, 10 MgSO$_4$, 5 Na-L-Ascorbate, 3 Myo-inositol, 3 Na Pyruvate, 2 Thiourea, 1.25 NaH$_2$PO$_4$·H2O, 0.5 CaCl$_2$, and 0.01 taurine (aCSF 1). All aCSF solutions were bubbled with carbogen (95% O$_2$; 5% CO$_2$).

Acute parasagittal slices (350 μm; maximum thickness for which healthy slices could be obtained) containing primary visual cortex from the right hemisphere were prepared with a Compresstome (Precisionary Instruments) in ice-cold aCSF 1 solution at a slice angle of 17° relative to the sagittal plane in order to preserve pyramidal cell apical dendrites. Slice angle was based on a Laplacian analysis of the Allen Mouse Common Coordinate Framework (Mouse CCF; http://help.brain-map.org/display/celltypes/Documentation) to determine the angle that maximally preserved cell arbors. Slices were recovered for 10 min in a holding chamber (BSK 12, Scientific Systems Design) containing oxygenated aCSF 1 maintained at 34°C (*Ting et al., 2014*; *Hájos and Mody, 2009*). After recovery, slices were kept in room temperature oxygenated aCSF holding solution (aCSF 2) containing (in mM): 94 NaCl, 25 D-Glucose, 25 NaHCO$_3$, 14 HEPES, 12.3 N-acetylcysteine, 5 Na-L-Ascorbate, 3 Myo-inositol, 3 Na Pyruvate, 2.5 KCl, 2 CaCl$_2$, 2 MgSO$_4$, 2 Thiourea, 1.25 NaH$_2$PO$_4$ • H$_2$O, 0.01 Taurine for a minimum of one hour prior to recording.

Human tissue surgically resected from adult cortex was obtained from patients undergoing neurosurgical procedures for the treatment of symptoms associated with epilepsy or tumor. Data were collected from 67 total slices from 22 surgical cases (17 epilepsy, 5 tumor, mean age ± SD 40 ± 17 years; min: 18, max: 75). Tissue obtained from surgery was distal to the core pathological tissue and

was deemed by the physician not to be of diagnostic value. Specimens were derived from the temporal lobe (13 epilepsy, 4 tumor) and the frontal lobe (4 epilepsy, 1 tumor). Specimens were placed in a sterile container filled with prechilled (2–4°C) aCSF 3 containing decreased sodium replaced with NMDG to reduce oxidative damage (*Zhao et al., 2011*) composed of (in mM): 92 NMDG, 2.5 KCl, 1.25 NaH$_2$PO$_4$, 30 NaHCO$_3$, 20 HEPES, 25 glucose, two thiourea, 5 Na-ascorbate, 3 Na-pyruvate, 0.5 CaCl$_2$ • 4H$_2$O and 10 MgSO$_4$ • 7H2O. pH was titrated to 7.3–7.4 with HCl and the osmolality was 300–305 mOsmoles/Kg. Although it has been noted that human CSF has a lower osmolality than that of mouse (*Bourque, 2008*), the solutions chosen here were previously optimized to promote tissue stability during dissection. Surgical specimens were transported (10–40 min) from the surgical site to the laboratory while continuously bubbled with carbogen.

Resected human tissue specimens were trimmed to isolate specific regions of interest, and larger specimens were cut into multiple pieces before trimming. Specimens were trimmed and mounted in order to best preserve intact cortical columns (spanning pial surface to white matter) before being sliced in aCSF 3 using a Compresstome. Slices were then transferred to oxygenated aCSF 3 maintained at 34°C for 10 min. Slices were kept in room temperature oxygenated aCSF holding solution (aCSF 4) containing, in mM: 92 NaCl, 30 NaHCO$_3$, 25 D-Glucose, 20 HEPES, 5 Na-L-Ascorbate, 3 Na Pyruvate, 2.5 KCl, 2 CaCl$_2$, 2 MgSO$_4$, 2 Thiourea, 1.2 NaH$_2$PO$_4$·H$_2$O for a minimum of one hour prior to recording.

## Electrophysiological recordings

Recording slices from mouse and human tissue were processed in largely the same manner, with a key difference being the external calcium concentration used for recording. Human slices were held in aCSF containing 1.3 mM calcium while mouse slices utilized 2.0 mM calcium. Below we discuss the full preparation for slice processing as well as the rationale for this calcium difference.

Slices were transferred to custom recording chambers perfused (2 mL/min) with aCSF maintained at 31–33°C, pH 7.2–7.3, and 30–50% oxygen saturation (as measured in the recording chamber). aCSF (aCSF 5) containing (in mM), 1.3 or 2 CaCl$_2$ (2.0 in mouse experiments and either 1.3 or 2.0 in human experiments), 12.5 D-Glucose, 1 or 2 MgSO$_4$, 1.25 NaH$_2$PO$_4$·H$_2$O, 3 KCl, 18 NaHCO$_3$, 126 NaCl, 0.16 Na L-Ascorbate. The concentration of calcium in the external recording solution, [Ca$^{++}$]$_e$, affects release probability and other aspects of synaptic dynamics (*Borst, 2010*; *Pala and Petersen, 2015*; *Jouhanneau et al., 2015*; *Urban-Ciecko et al., 2015*). Although [Ca$^{++}$]$_e$ concentrations close to 1 mM are expected to most closely approximate in-vivo-like synaptic dynamics, most prior multi-patch studies used elevated calcium concentrations to increase the amplitude of EPSPs and improve throughput. In our mouse recordings, we used 2.0 mM [Ca$^{++}$]$_e$ to be consistent with previous connectivity studies (*Markram, 1997*; *Reyes and Sakmann, 1999*; *Perin et al., 2011*; *Jiang et al., 2015*) and to help ensure the success of our system integration test. We selected 1.3 mM [Ca$^{++}$]$_e$ for our human experiments because of reports that the synaptic strength is higher than in mouse and to minimize the complex events that can be initiated by individual spikes in human tissue (*Molnár et al., 2008*) that make identifying monosynaptic connectivity challenging.

Slices were visualized using oblique (Olympus; WI-OBCD) infrared illumination using 40x or 4x objectives (Olympus) on a custom motorized stage (Scientifica), and images were captured using a digital sCMOS camera (Hamamatsu; Flash 4.0 V2). Pipette positioning, imaging, and subsequent image analysis were performed using the python platform acq4 (acq4.org, *Campagnola et al., 2014*). Eight electrode headstages were arranged around the recording chamber, fitted with custom headstage shields to reduce crosstalk artifacts, and independently controlled using modified triple axis motors (Scientifica; PatchStar). Signals were amplified using Multiclamp 700B amplifiers (Molecular Devices) and digitized at 50–200 kHz using ITC-1600 digitizers (Heka). Pipette pressure was controlled using electro-pneumatic pressure control valves (Proportion-Air; PA2193) and manually applied mouth pressure.

Recording pipettes were pulled from thick-walled filamented borosilicate glass (Sutter Instruments) using a DMZ Zeitz-Puller (Zeitz) to a tip resistance of 3–8 MΩ (diameter ~ 1.25 μm), and filled with internal solution containing (in mM): 130 K-gluconate, 10 HEPES, 0.3 ethylene glycol-bis(β-aminoethyl ether)-N,N,N′,N′-tetraacetic acid (EGTA), 3 KCl, 0.23 Na$_2$GTP, 6.35 Na$_2$Phosphocreatine, 3.4 Mg-ATP, 13.4 Biocytin, and either 25 μM Alexa-594 (excited at 880 nm) for optogenetic experiments, 50 μM Cascade Blue dye (excited at 490 nm), or 50 μM Alexa-488 (excited at 565 nm). Osmolarity was between 280 and 295 mOsm titrated with ~ 4 mM sucrose, pH between 7.2 and 7.3

titrated with ~ 10 mM KOH. The liquid junction potential between our internal solution and aCSF five was measured to be 9.40 mV ± 0.59; SD. All electrophysiological values are reported without junction potential correction.

In experiments on human tissue and wild-type mice, clusters of up to eight excitatory neurons were selected based on cortical layer, somatic appearance, and depth from the surface of the slice; Neurons deep in the tissue (depth from slice surface ≥ 40 µm; *Figure 4—figure supplement 2D*) were patched with the assistance of automated pipette control. Tissue distortion and damage was minimized by moving through the tissue on a trajectory that was collinear with long axis of the pipette with minimal positive pressure. In transgenic mice, cells were also targeted based on fluorescent reporter expression. All cells were confirmed as excitatory post-experiment either by their EPSPs onto other recorded neurons (*Figure 1—figure supplement 1A*) or by their pyramidal morphology, visualized using either biocytin (*Figure 1A*) or fluorescent dye from the pipette (*Figure 1B*). Cell intrinsic fluorescence was confirmed post-hoc via manual inspection of image stacks to evaluate signal overlap of the transgenic fluorescent reporter and the fluorescent dye introduced via pipettes (*Figure 1B*). Whole-cell patch clamp electrophysiological recordings were performed at −70 mV to preferentially measure excitatory inputs. Resting membrane potential was maintained within 2 mV of −70 mV using automated bias current injection during the inter-trial interval. Custom software, Multi-channel Igor Electrophysiology Suite (MIES; https://github.com/AllenInstitute/MIES), written in Igor Pro (WaveMetrics), was used for data acquisition and pipette pressure regulation. A brief, 10 ms test pulse was used to monitor access (24.18 ± 9.24 MΩ) and input (166.59 ± 84.0 MΩ) resistance over the duration of the recording. During recordings, cells were stimulated using brief current injections (1.5 or 3 ms) to drive trains of 12 action potentials (*Figure 1—figure supplement 1A*) at frequencies of 10, 20, 50, or 100 Hz to induce short-term plasticity (STP). Stimulus amplitudes (mean ± SD, 1.2 ± 0.5 nA) were adjusted to ensure spiking on every pulse across a range of cell types and frequencies. A delay period inserted between the 8th and 9th pulses allowed testing of recovery from STP. In most recordings this delay period was 250 ms; for 50 Hz stimulation, longer delay periods (500, 1000, 2000, and 4000 ms) were used as well (see *Figure 5A*). Connectivity was first evaluated in voltage clamp (holding at −70 mV), prior to entering the experimental workflow shown in *Figure 1—figure supplement 1C–G* in the current clamp recording configuration. Individual recordings were assessed against standardized quality control metrics in order to be included in each subsequent analysis (see *Figure 1—figure supplement 1C–G* and *Table 1*). Experimental protocols were repeated five times for each stimulation frequency and delay interval. Stimuli were interleaved between cells such that only one cell was spiking at a time, and no two cells were ever evoked to spike within 150 ms of each other.

## Data analysis

Postsynaptic recording traces were aligned to the time of the presynaptic spike evoked from the stimuli described above (*Figure 1—figure supplement 1B*). Postsynaptic potentials (PSPs) were identified by manual inspection of spike-aligned and averaged recordings in response to evoked spikes, as well as a parallel inspection of background noise. A classifier (described below) was later used to highlight possible identification errors, which were then manually corrected. Connection probabilities within 100 µm intersomatic distance were compared between cell types using Fisher's exact test of 2 × 2 contingency tables (connected, unconnected). The relationship between connectivity and intersomatic distance (measured from 3D cell positions) was analyzed by binning connections in 40 µm windows and calculating the 95% Jeffreys Bayesian confidence interval for each bin.

Subsets of the 131 mouse and 76 human connections found in this study were analyzed for strength, kinetics, and STP based on specific quality control criteria *Figure 1—figure supplement 1C–G*, *Table 1*). EPSP strength, kinetics, and coefficient of variance (CV) measurements (*Figures 1* and *3*) were conducted on the first-pulse response from 10, 20, and 50 Hz stimulation trains which were time-aligned to the presynaptic spike and averaged for each connection. Connections were included for strength and kinetics analysis according to the analysis flowchart in *Figure 1—figure supplement 1E and F*. Briefly, the postsynaptic cell had an auto bias current less than 800 pA (mean bias current −95 ± 182 pA), there was no spontaneous spiking, the stimulus artifact was minimal (<30 µV), and the PSP was positive. Individual recording sweeps were included if the baseline potential drift was smaller than ±5 mV from holding (−70 mV) and the mean baseline 10 ms preceding stimulation was less than three standard deviations of the mean baseline across sweeps. In the QC

passed data, strength and kinetics were measured from a double exponential fit that approximates the shape of the PSP:

$$y(t) = A \left(1 - e^{-(t-t_0)/\tau_r}\right)^2 * e^{-(t-t_0)/\tau_d} \tag{1}$$

Best fit parameters were obtained using the Non-Linear Least-Squares Minimization and Curve-Fitting package for Python (LMFIT; *Newville and Allen, 2014*). To improve the quality of fitting, the root mean square error was weighted (WRMSE) differently throughout the trace. The rising phase of the PSP was most heavily weighted, the baseline and decay regions were intermediately weighted, and the region of the presynaptic stimulus, which often contained crosstalk artifacts was masked. Amplitude was measured as the peak of the PSP fit (*Figure 1—figure supplement 1B*). Kinetics were measured from connections in which the WRMSE of the fit was less than 8. Latency is reported as the duration from the point of maximum *dV/dt* in the presynaptic spike until the foot of the PSP (*Figure 1—figure supplement 1B*), taken from the x-offset in the double exponential fit. Rise time is reported as the duration from 20% of the peak until 80% of the peak of the PSP (*Figure 1—figure supplement 1B*). Significance of differences in PSP amplitude, latency, and rise time across layers or Cre-lines were assessed with a Kruskal-Wallis test.

STP (*Figures 5* and *6*) was measured from a similar subset of connections that included the quality control criteria above and also excluded responses smaller than 0.5 mV in amplitude to minimize the effect of noise on mean response which might impact the model (*Figure 1—figure supplement 1G*, *Equations 4 and 5*). Connections or individual sweeps that had a baseline holding potential of −55 mV (±5 mV) were reintroduced for this analysis if they met the QC criteria. Normalized PSP amplitudes (relative to the first pulse) were estimated using an exponential deconvolution ($\tau$ = 15 ms; *Richardson and Silberberg, 2008*) to compensate for summation from prior PSPs and to increase signal-to-noise in measuring PSP amplitudes:

$$D(t) = V + \tau \frac{dV}{dt} \tag{2}$$

Although the fixed deconvolution time constant of 15 ms may differ significantly from the actual time constant of each cell, in practice this has little effect on the normalized amplitudes used in STP measurements (for example, using this time constant to measure amplitudes from a simulated 100 Hz train with a cell time constant of 30 ms only resulted in 3% error in the measurement of PSP amplitudes relative to the first pulse; data not shown). The peak amplitudes from the deconvolved traces were used to measure the change in response magnitude over the course of stimulus trains. We measured the magnitude of short-term depression or facilitation using the ratio between the first and last (eighth) pulses in an induction pulse train, whereas recovery from depression or facilitation was measured by the ratio between the first pulse and the ninth pulse, which followed a recovery delay. Kruskal-Wallis tests were used to assess significance of STP between multiple layers. A descriptive model was used to capture features of short-term depression in Rorb, Sim1, and Tlx3 connections (*Equations 4 and 5*).

## Automatic synapse detection

To aid in the detection of synaptic connections, a support vector machine classifier (implemented with the 'sklearn' python package, *Pedregosa et al., 2012*) was trained to discriminate between experiments in which EPSPs were either visible or not visible to a human annotator. The classifier required a diverse set of features (*Supplementary File 1*) that were pre-processed from the raw postsynaptic recordings immediately surrounding each evoked presynaptic spike. Averaged responses were characterized by curve fitting (*Equation 1*; *Figure 1—figure supplement 1B*) and the fit parameters as well as the normalized RMS error were provided as features to the classifier. Additionally, individual response recordings were analyzed by measuring the amplitude and time of the peak of each exponentially deconvolved response over a 3 ms window beginning 1 ms after the presynaptic spike, compared to a 10 ms window preceding the stimulus pulse (*Figure 1—figure supplement 1B*, bottom). Although these individual measurements were often noisy (average background RMS noise 607 ± 419 µV), their distribution over hundreds of trials could be compared to similar distributions measured from background noise (e.g. *Figure 4C*). Distributions were compared

using a Kolmogorov-Smirnov test (from the 'scipy.stats' Python package) and the p values were used as input features for the classifier.

After training on 1854 manually labeled examples, the classifier was tested against a withheld set of 2642 examples and achieved an overall accuracy of 95% (56/61 true positive, connected; 2457/2581 true negative, not connected). False positives and negatives were manually reassessed and the annotations corrected when appropriate. Whereas false negatives were usually the result of a classifier failure, false positives were frequently found to have been misclassified during the initial manual annotation.

## Analysis of synapse detection sensitivity

To measure the minimum detectable PSP size for each connection probed, artificial PSPs were added to recordings of background noise taken from the postsynaptic cell. PSPs were generated using *Equation 1* with a foot-to-peak rise time of 2 ms (except where specified in *Figure 3D*). PSP latencies were selected from a gaussian distribution centered at 2 ms with a 200 μs standard deviation. PSP amplitudes were generated by the product of two random variables: one binomially distributed (p=0.2, n = 24) to mimic stochastic vesicle release, and the other normally distributed (mean = 1, SD = 0.3) to account for differences in vesicle size and receptor efficacy. PSPs were then scaled uniformly to achieve a specific mean amplitude. The resulting simulated responses were qualitatively similar to typical synaptic responses encountered in our dataset, although they lacked the synapse-to-synapse variability in CV, due to the selection of fixed distribution parameters listed above.

For each connection probed, the number of simulated PSPs generated was the same as the number of presynaptic spikes elicited during the experiment. These PSPs were then fed through the same preprocessing and classification system that was used to detect synaptic connections in real data, and the classification probability was calculated from the classifier (using sklearn.svm.SVC.predict_proba). This process was repeated eight times (with PSPs generated randomly each time) and the average classification probability was recorded. This yields an estimate of whether a synaptic connection would be detected or overlooked, given the combination of sample count, background noise characteristics, and PSP strength and kinetics.

By repeating this process for several different values of mean PSP amplitude, we could identify, for each putative connection probed, a plausible minimum detectable PSP amplitude. This minimum detectable amplitude was defined as the PSP amplitude at which the classifier would detect the synapse in 50% of trials (interpolated from adjacent amplitudes).

## EPSP amplitude run-down over duration of experiment

The amplitude of the EPSPs initiated by the first pulse of the stimulus trains through-out the duration of the experiment were characterized by fitting the exponential fit (*Equation 1*) to individual EPSPs. Individual EPSPs are often noisy, thus, only connections where individual EPSPs looked well fit were used to assess rundown. In order to further discount variations in the measurements of individual responses, the run-down was characterized via a linear regression of EPSP amplitude versus time for sweeps with a holding potential between −75 and −65 mV. We observed run-down in all synapse types in percent per minute as follows: layer 2/3 to layer 2/3: median 1.9, average 3.7, std 3.8, Rorb to Rorb: median 4.0, average 3.9, std 0.81, Sim1 to Sim1: median 1.4, average 0.61, std 3.2, Tlx3 to Tlx: median 1.5, average 1.9, std 5.3.

## Theoretical synaptic modeling

Among the wealth of mechanisms described to contribute to short-term plasticity, we chose to focus on depression and use-dependent replenishment, as these models provide the best fits when corrected for the number of parameters used. The standard depression model (*Hennig, 2013*; *Mongillo et al., 2008*; *Richardson et al., 2005*) was not sufficient to account for changes during multiple stimulating frequencies, thus we included use-dependent replenishment. The PSP, $w = A \cdot n \cdot P_0$, is modeled as being proportional to the fraction of vesicles ($n$) and the release probability ($P_0$); the constant $A$ determines the strength of the connection. Synaptic depression was modelled via depletion of vesicles (*Hennig, 2013*; *Mongillo et al., 2008*; *Richardson et al., 2005*),

$$\frac{dn}{dt} = \frac{1-n}{\tau_r} - P_0 n\delta(t-t_k) \tag{4}$$

where $t_k$ is the time of presynaptic spike and $\tau_r$ is the time constant for vesicle replenishment. The speed of replenishment can vary over time depending on the history of presynaptic spikes, which can be captured by time constant $\tau_r$ evolving according to *Equation 5* (*Fuhrmann et al., 2002*; *Hennig, 2013*),

$$\frac{d\tau_r}{dt} = \frac{\tau_{r0} - \tau_r}{\tau_{FDR}} - a_{FDR}\tau_r\delta(t-t_k) \tag{5}$$

where $\tau_{FDR}$ is the time constant of use-dependent replenishment, $a_{FDR}$ represents the amount of updates elicited by a presynaptic spike and $\tau_{r0}$ is the baseline time constant.

For Rorb, Sim1, and Tlx3 synapses, we optimized the parameters ($P_0$, $\tau_{r0}$, $\tau_{FDR}$ and $a_{FDR}$) to account for time courses of PSPs. To characterize the short-term synaptic plasticity of synapse classes, we averaged PSPs over all available synapses depending on stimulation frequencies and delays between 8th and 9th presynaptic pulses and fitted to the model. We used LMFIT (*Newville and Allen, 2014*) to perform non-linear least-square minimization and report the optimal values and standard errors estimated from the covariance matrix.

Paired Z-scores for $P_0$ and $\tau_{r0}$ were calculated from the standard error returned during parameter optimization according to *Equation 6*,

$$Z-score = \frac{|X_1 - X_2|}{\sqrt{SE_1^2 + SE_2^2}} \tag{6}$$

where $X_1$ and $X_2$ are $P_0$ or $\tau_{r0}$ for two groups and their associated standard error.

## Histology and morphology

After completing electrophysiological recordings, slices were transferred from the recording chamber and fixed in solution containing 4% PFA and 2.5% glutaraldehyde for 2 days (>40 hr) at 4°C. After fixation, slices were transferred and washed in phosphate buffer saline (PBS) solution for 1–7 days.

Sections were processed using 3,3′-diaminobenzidine (DAB) peroxidase substrate kit to identify recorded neurons filled with biocytin. Free floating sections were first incubated with 5 µM 4′,6-diamidino-2-phenylindole (DAPI) in PBS for 15 min at room temperature and then triple washed in PBS (3 × 10 min). Sections were transferred to a 1% $H_2O_2$ (in PBS) for 30 min and then triple washed in PBS. A DAB substrate kit (VectorLabs) was used to stain for neurons filled with biocytin. Sections were mounted on gelatin-coated slides and coverslipped with Aqua-Poly/Mount (Polysciences).

Slides were imaged on an AxioImager Z2 microscope (Zeiss) equipped with an Axiocam 506 camera (Zeiss) and acquired via the Zeiss Efficient Navigation software. Tiled mosaic images of whole slices were acquired via automated scanning and stitching of several 20X images to generate both biocytin-labeled images (used to assess cell morphology) and DAPI-labeled images (used to identify cortical layer boundaries) of the entire slice.

## Two-photon optogenetic experiments

Connectivity mapping experiments were performed on a two-photon laser scanning microscope (Bruker Corp) with a tunable pulsed Ti:Sapphire laser (Chameleon Ultra, Coherent) for imaging, and a fixed wavelength (1060 nm) pulsed laser (Fidelity Femtosecond, Coherent) for stimulation. A 63x, 1.0 NA water immersion objective (Zeiss) was used for all experiments. Two-photon images were acquired with PrairieView software (Bruker Corp), and stimulation targets were manually placed on these reference images to target ReaChR-positive cells. Photoactivation stimuli were triggered by a TTL pulse generated within MIES acquisition software. The voltage output controlling the photoactivation Pockels cell was recorded within MIES for post-hoc alignment of physiological recordings with the timing of photoactivation. To characterize the effectiveness and specificity of stimulation parameters, we made loose seal recordings on to EYFP/ReaChR-labelled neurons (*Figure 4—figure supplement 2A*). For all data presented here, the photostimulation pattern consisted of a spiral 5 µm in diameter with five revolutions traced over a 25 ms duration. We first determined the minimum light

power necessary to evoke reliable firing of action potentials. This minimum power varied across cells from 2.6 to 20.3 mW (*Figure 4—figure supplement 2B*). A photostimulus of 18 mW intensity was sufficient to evoke spiking in 92% of cells tested (12/13 cells). The average latency of firing at this power was 12.9 ± 6.1 ms and the associated jitter was 0.98 ± 0.58 ms (*Figure 4—figure supplement 2C,D*).

Within the same experiments, we characterized the spatial specificity of these stimulation parameters. First, to determine the probability of off-target photoactivation of cells within the same focal plane, we delivered stimuli in a radial grid pattern containing seven spokes with stimuli spaced 10, 20 and 30 μm away from the center of the recorded cell ((*Figure 4—figure supplement 2E*). Spike probability fell to 0.5 at a lateral distance of 12.0 μm. Finally, we determined the axial resolution of our photoactivation paradigm by offsetting the focus of the objective relative to the recorded cell. Consistent with previous studies (*Packer et al., 2012*; *Prakash et al., 2012*), axial resolution was inferior compared to lateral resolution (spike probability = 0.5 at 26.7 μm) but was still near cellular resolution ((*Figure 4—figure supplement 2F*).

For two-photon mapping experiments, 1–2 neurons were patched and membrane potential was maintained near −70 μV with auto bias current injection. Neurons were filled with 50 μM Alexa-594 to visualize cell morphology ((*Figure 4—figure supplement 1A*). The orientation of the apical dendrite was utilized to align photostimulation sites across experiments in downstream analyses. Each putative presynaptic neuron was stimulated 10–20 times, with the parameters described above. Photostimulation was performed in 'rounds' during which EYFP-labelled neurons within a single field of view were sequentially targeted (3–12 neurons/round). Stimulation protocols were constrained such that the inter-stimulus interval between neurons was ≥ 2 s and the inter-stimulus interval for a given neuron was ≥ 10 s.

Importantly, the photostimulus utilized in our mapping experiments generated spiking in most - but not all – Tlx3 neurons tested (92% spike probability; (*Figure 4—figure supplement 2*). To estimate a false negative rate associated with incomplete photosensitivity, we utilized the multipatch dataset to establish a prior probability of Tlx3 recurrent connectivity (4.74%; *Table 1*). Assuming connection probability to be independent of photosensitivity, we estimate the false negative rate as

(1-photosensitivity)*prior probability of connectivity = (1–0.92)*0.0474 = 0.0038 or 0.38%.

Notably, this low false negative rate is partly due to the observed sparsity of recurrent connectivity between Tlx3-positive neurons. Higher error rates are to be expected when using two-photon optogenetics to probe high-probability synaptic connections.

Photostimulus responses were scored as connection, no connection or as containing a direct stimulation artifact by manual annotation. To assist in these user-generated calls, we incorporated a signal-to-noise measure for our optogenetic mapping data. Current clamp traces were low pass filtered at 1 kHz and baseline subtracted. The voltage-deconvolution technique (*Equation 2*) was then applied. The value of τ was set between 10 and 40 ms. Deconvolved traces were high-pass filtered at 30 Hz, and peaks larger than three standard deviations above pre-stimulus baseline were used for further analysis ((*Figure 4—figure supplement 1B,C*). We then measured the number of peaks in both 'signal' and 'noise' regions. The 'signal' region was a 100 ms window 5–105 ms after the onset of the photostimulus, and the 'noise' region was a 100 ms window 145–45 ms before the stimulus onset. To compensate for jitter known to be present in two-photon mediated stimulation, we determined a 10 ms subset within each 100 ms window that gave the maximum number of unique trials containing threshold-crossing events. The median of the peak within this 10 ms window was found across all trials in both 'signal' and 'noise' regions, and the mean of a 25 ms window preceding both regions was subtracted to produce our final signal and noise values.

We plotted the signal against the noise for all stimulus locations ((*Figure 4—figure supplement 1D*), and found that most points with a high signal-to-noise ratio contained either a synaptic response or an artifact produced by direct stimulation of the recorded (opsin-expressing) cell. 88% (15/17) manually identified connections had a signal to noise ratio > 1.5 ((*Figure 4—figure supplement 1E*). By contrast, the same was true of only 1.8% (31/1720) of cells scored as 'no connection'. Therefore, our signal-to-noise analyses highlight quantitatively distinct features of our connection calls. The presence of direct stimulation artifacts prevents us from unambiguously identifying synaptic connections between nearby neurons. Therefore, when estimating connection probability by two-photon optogenetics, we did not include putative presynaptic cells within 50 μm of the recorded neuron where the direct stimulation artifact was largest (*Figure 4F*). It is also important to consider

differences in detection limits between multipatch and two-photon optogenetics. The amplitudes of EPSPs from optogenetically identified connections ranged from 0.10 to 0.78 mV (mean amplitude 0.30 ± 0.21; SD). Multipatch recordings allowed spike-aligned averaging over many trials and resulted in detection of synaptic responses < 0.10 mV (*Figure 1F*). Therefore, while two-photon optogenetics provides dense sampling over large inter-somatic distances, estimates of connectivity are likely biased toward large and reliable synaptic connections.

## Acknowledgements

The authors thank the Allen Institute founder, Paul G Allen, for his vision, encouragement and support. Thank you to the Allen Institute for Brain Science Tissue Processing, Histology, Imaging, and Morphology and 3D reconstruction teams for preparing and processing mouse tissue. We further thank the Tissue Procurement, Tissue Processing, and Facilities teams for help in coordinating the logistics of human surgical tissue collection, transport, and processing. We are also grateful to our collaborators at the local hospital sites, including Tracie Granger, Caryl Tongco, Matt Ormond, Jae-Guen Yoon, Nathan Hansen, Niki Ellington, Rachel Iverson (Swedish Medical Center), Carolyn Bea, Gina DeNoble and Allison Beller (Harborview Medical Center). We thank Dirk C Keene at Harborview/UW for consultation and support. This work was supported by the Allen Institute for Brain Science, National Institutes of Health grant U01MH105982 to HZ, and the Howard Hughes Medical Institute grant to GJM.

## Additional information

### Funding

| Funder | Grant reference number | Author |
|---|---|---|
| National Institutes of Health | U01MH105982 | Luke Campagnola<br>Hongkui Zeng |
| Howard Hughes Medical Institute | | Gabe Murphy |

The funders had no role in study design, data collection and interpretation, or the decision to submit the work for publication.

### Author contributions

Stephanie C Seeman, Data curation, Software, Formal analysis, Visualization, Writing—original draft, Project administration, Writing—review and editing; Luke Campagnola, Data curation, Software, Formal analysis, Validation, Investigation, Visualization, Writing—original draft, Writing—review and editing; Pasha A Davoudian, Formal analysis, Investigation, Visualization, Writing—review and editing; Alex Hoggarth, Formal analysis, Investigation, Visualization, Writing—original draft, Writing—review and editing; Travis A Hage, Data curation, Formal analysis, Validation, Investigation, Visualization; Alice Bosma-Moody, Data curation, Software, Formal analysis, Investigation, Methodology; Christopher A Baker, Data curation, Formal analysis, Investigation, Methodology; Jung Hoon Lee, Stefan Mihalas, Computational modeling; Corinne Teeter, Software, Formal analysis; Andrew L Ko, Jeffrey G Ojemann, Ryder P Gwinn, Daniel L Silbergeld, Charles Cobbs, Resources, Neurosurgery; John Phillips, Conceptualization, Resources, Supervision; Ed Lein, Resources, Funding acquisition, Project administration; Gabe Murphy, Resources, Supervision, Methodology, Project administration, Writing—review and editing; Christof Koch, Conceptualization, Resources, Writing—review and editing; Hongkui Zeng, Conceptualization, Resources, Supervision, Funding acquisition, Writing—review and editing; Tim Jarsky, Conceptualization, Resources, Data curation, Software, Formal analysis, Supervision, Investigation, Visualization, Methodology, Writing—original draft, Project administration, Writing—review and editing

### Author ORCIDs

Christopher A Baker http://orcid.org/0000-0002-0604-8449
Stefan Mihalas http://orcid.org/0000-0002-2629-7100

Ed Lein [ID] http://orcid.org/0000-0001-9012-6552
Hongkui Zeng [ID] http://orcid.org/0000-0002-0326-5878
Tim Jarsky [ID] http://orcid.org/0000-0002-4399-539X

## Ethics

Human subjects: All patients provided informed consent for tissue donation, and all experimental uses were approved by the respective hospital Institutional Review Board before commencing the study (the Swedish Institutional Review Board for patients at Swedish Neuroscience Institute [1] or the Institutional Review Board of the University of Washington for patients at Harborview Medical Center [2]). All experiments and methods were performed in accordance with the relevant guidelines and regulations. [1] Swedish Dr. Cobb's protocol #1111798 Title: Brain Tumor Biomarker Database Dr. Gwinn's protocol #1068035 / Title: Tissue Procurement and Repository Protocol for Epilepsy Tissue* [2] University of Washington (Harborview) #HSD No. 49119 / Title: Cellular and Circuit Organization of Human Brain Collaboration Allen Institute; Ed Lein - Determination of Not Human Subjects / Title: Cellular and Circuit Organization of Human Brain Project

Animal experimentation: Mice used in this study were housed and sacrificed according to protocols approved by the Institutional Care and Use Committee at the Allen Institute (Seattle, WA), in accordance with the National Institutes of Health guidelines.

## Decision letter and Author response

Decision letter https://doi.org/10.7554/eLife.37349.031
Author response https://doi.org/10.7554/eLife.37349.032

## Additional files

### Supplementary files

• Supplementary file 1. Description of features extracted from raw data to use in synapse classifier.
DOI: https://doi.org/10.7554/eLife.37349.028

• Transparent reporting form
DOI: https://doi.org/10.7554/eLife.37349.029

### Data availability

Processed data (CSV files) has been uploaded with the manuscript for each figure. Raw Neurodata Without Borders (NWB) files are too large (>1TB) to accompany the manuscript but will be made available through brain-map.org with our first data release (scheduled for 2019). We have not uploaded the NWB files to a data repository because we want a single point of access for our data. In the interim period before public release, NWB files will be made available by request to timj@alleninstitute.org Analysis code can be found at the following links: https://github.com/AllenInstitute/multipatch_analysis (copy archived at https://github.com/elifesciences-publications/multipatch_analysis) https://github.com/AllenInstitute/neuroanalysis (copy archived at https://github.com/elifesciences-publications/neuroanalysis).

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
