## [Decision Letter]

Thank you for submitting your article "Sparse recurrent excitatory connectivity in the microcircuit of the adult mouse and human cortex" for consideration by *eLife*. Your article has been reviewed by Eve Marder as the Senior Editor, a Reviewing Editor, and three reviewers.

The reviewers have discussed the reviews with one another and the Reviewing Editor has drafted this decision to help you prepare a revised submission. We understand that this manuscript represents a good deal of work, but also feel that requires significant editorial attention to make it self-consistent and transparent.

Summary:

Seeman et al., have explored visual cortex excitatory connectivity across multiple layers and cell types using acute slices, patch-clamp recordings, 2-photon optogenetic activation, and computer simulations. The authors explore both mouse and human connectivity, although only specifically visual cortex in the rodent, not human. Human tissue was resected from different brain regions of patients with epilepsy or tumours. The authors find that connectivity is higher and short-term depression lower in supragranular layers. The authors also find that connective strength in human tissue is stronger, as previously shown.

The manuscript is well written and the figures are largely quite clear. The novelty of these results lies in the combination of studying several cortical layers simultaneously in mouse as well as human tissue. The authors should address all the major concerns raised by the reviewers by either extending and strengthening the weaker sections or by removing the weakest data, and then fairly and completely discussing the methodology for what remains. We remind you that *eLife* also publishes Research Advances that would, in principle, allow you to follow this study with one that strengthens the weaker aspects of what is now here, should you wish to remove them, as might be advisable.

Essential revisions:

1) Developing a standardized pipeline for analyzing the synaptic connectivity of the cortex and other brain regions would represent an advance in the field. However, in many places, the authors do not justify their choices for their experimental procedures. How slices are cut may affect connectivity rates. For example, studies in slices from older mice have used an NMDG-based solution, different from the one used in the current study (Jiang et al., 2015) as well as common alternatives including kynurenic acid- and sucrose-based solutions (see, for example, Galaretta and Hestrin, 2002 (2-7 months old); Crandall et al., 2017). When developing this standardized pipeline, what was the justification for adopting the experimental procedures for preparing the mouse and human slices? Were comparisons made among different approaches including comparisons of slice thickness, slice orientation, depth of recording, and so on that may affect connectivity rates? These variables should be assessed before settling on a standardized pipeline. Furthermore, the authors do not include enough experimental detail to enable the reader to evaluate the standardized pipeline and compare it to prior studies (see below).

2) This finding in L6 PCs that connectivity is low is not so strange given the prior literature, e.g. Velez-Fort et al., 2014. The authors genetically target the less connected form of L6 PC and then report that they cannot find many connections, but this is expected when targeting CT L6 neurons. The most rigorous way to address this would probably be to additionally target and record from L6 PCs other than the ones that express Ntsr1. Alternatively, it would be possible to take out all data on L6, because it is presently quite limited anyway.

3) Technically our major concern lies with the composition of the whole-cell recording solution used. The authors use an internal containing 0.3 mM EGTA. Previous work on unitary excitatory output of layer 2/3 pyramidal neurons has demonstrated that this concentration has a major influence on uEPSP amplitude and dynamics in a target specific manner (Rozov et al., 2001) – decreasing use-dependent facilitation. It is possible that the use of EGTA in the internal solution has flattened the disparity between the use-dependent dynamics of in each layer, and also may have implications for comparison across species. This is a major problem that requires addressing.

4) It has been shown that connectivity depends on the pre- and post-synaptic cell type (see, for example, Brown and Hestrin. 2009). However, here the authors are likely mixing synapse types in their analyses. Cre is expressed in more than one cell type in the Ntsr1-Cre line (Chevée et al., 2018, Tasic et al., 2016) and the Tlx3-Cre line (Tasic et a.l, 2017). Layer 2/3 includes more than one cell type (see, for example, Yamashita et al., 2013; Yamashita et al., 2018; Tasic et al., 2016; Tasic et al., 2017). Thus, the connectivity rates and the average synaptic properties shown here may not represent connectivity rates among specific cell types but rather represent mixtures of synapse types within a layer.

5) The machine learning approach is basically a form of advanced template matching that looks for an event with the right properties. But at no stage can we see that the machine-learning algorithm knows exactly where the presynaptic spike is. Detection should explicitly model the fact that the time of the action potential is known, since it would strengthen its performance, in turn affecting estimates.

6) Figure 4EF data set is small, and the resulting connectivity rate in panel F is low compared to the prior literature (e.g. Perin, Berger and Markram, 2011). Either this n should be increased, or maybe simply remove panels E-F?

7) What about bidirectional connections? Both Song, 2005 and Lefort, 2009 make a big deal of those, it would seem strange not to mention them.

8) E.g. albino animals are known to have different connectivity patterns, so if some of these transgenic lines (Materials and methods section, etc.) were created on an albino background, then please discuss. Differences attributed to layers could in reality be due to the transgenic mouse line used.

---

## [Author Response]

Summary:Seeman et al. have explored visual cortex excitatory connectivity across multiple layers and cell types using acute slices, patch-clamp recordings, 2-photon optogenetic activation, and computer simulations. The authors explore both mouse and human connectivity, although only specifically visual cortex in the rodent, not human. Human tissue was resected from different brain regions of patients with epilepsy or tumours. The authors find that connectivity is higher and short-term depression lower in supragranular layers. The authors also find that connective strength in human tissue is stronger, as previously shown.The manuscript is well written and the figures are largely quite clear. The novelty of these results lies in the combination of studying several cortical layers simultaneously in mouse as well as human tissue. The authors should address all the major concerns raised by the reviewers by either extending and strengthening the weaker sections or by removing the weakest data, and then fairly and completely discussing the methodology for what remains. We remind you that eLife also published Research Advances that would, in principle, allow you to follow this study with one that strengthens the weaker aspects of what is now here, should you wish to remove them, as might be advisable.

In this revision we have strengthened a number of the weaker sections by adding new data (see comment responses for details). In the time available, we were unable to add recurrent connections in layer 6 CC neurons. We will consider adding this data to an *eLife* Research Advance once the experiments are complete. As suggested, we removed the calcium exchange experiments.

Essential revisions:1) Developing a standardized pipeline for analyzing the synaptic connectivity of the cortex and other brain regions would represent an advance in the field. However, in many places, the authors do not justify their choices for their experimental procedures. How slices are cut may affect connectivity rates. For example, studies in slices from older mice have used an NMDG-based solution, different from the one used in the current study (Jiang et al., 2015) as well as common alternatives including kynurenic acid- and sucrose-based solutions (see, for example, Galaretta and Hestrin, 2002 (2-7 months old); Crandall et al., 2017). When developing this standardized pipeline, what was the justification for adopting the experimental procedures for preparing the mouse and human slices? Were comparisons made among different approaches including comparisons of slice thickness, slice orientation, depth of recording, and so on that may affect connectivity rates? These variables should be assessed before settling on a standardized pipeline. Furthermore, the authors do not include enough experimental detail to enable the reader to evaluate the standardized pipeline and compare it to prior studies (see below).

We’ve enhanced the existing justification for methodological choices with text, references, and supplementary figures. Please let us know if there are additional areas where the manuscript would benefit from further detail. Also, please consider that the data reported here are not production pipeline data. A purpose of the integration test is to assess methods and the methodological analysis is ongoing. We will share the results of the methodological analysis in a white paper (our standard practice) as well as in a pipeline platform paper in the future. A motivating factor for publishing integration test data was to expose our project to early peer review and open it to the scientific community for feedback. We are delighted with the detailed and helpful reviewer comments. Thank you.

In regards to specific areas of concern mentioned in the comment:

Our slicing conditions were the same conditions as our Cell-Types pipeline (Gouwens et al., 2018). The methods were based on Hajos and Mody, (2009) and Ting et al., (2014).

We’ve added an analysis of the effect of slice location and recording depth to the manuscript (Figure 4—figure supplement 1).

2) This finding in L6 PCs that connectivity is low is not so strange given the prior literature, e.g. Velez-Fort et al., 2014. The authors genetically target the less connected form of L6 PC and then report that they cannot find many connections, but this is expected when targeting CT L6 neurons. The most rigorous way to address this would probably be to additionally target and record from L6 PCs other than the ones that express Ntsr1. Alternatively, it would be possible to take out all data on L6, because it is presently quite limited anyway.

As suggested, we've added references to West et al., 2006 and Velez-Fort et al., 2014 and drawn attention to the fact that recurrent connectivity is more common for L6 CC neurons than L6 CT (Ntsr1) neurons.

We continue to favor including the Ntsr1 data for several reasons: (1) It is the first time connectivity versus distance has been quantified for this cell class (2) The use of high SNR whole-cell recordings rather than sharp electrodes. (3) Our work is the first functional characterization of recurrent connectivity for this mouse line in the adult mouse primary visual cortex. (4) Although the low number of NTSR1 synapses we have recorded from prevents an analysis of synaptic properties, the large number of connections probed yields high confidence in our measurement of connectivity. Accurate estimates of connectivity as a function of distance are a critical component of network models.

We've begun to take the more rigorous approach suggested, but we ran out of time; non-fluorescent cells require morphological calls, which have lower throughput and adds weeks onto the processing pipeline. We plan to include this data in an *eLife* Research Advance.

3) Technically our major concern lies with the composition of the whole-cell recording solution used. The authors use an internal containing 0.3 mM EGTA. Previous work on unitary excitatory output of layer 2/3 pyramidal neurons has demonstrated that this concentration has a major influence on uEPSP amplitude and dynamics in a target specific manner (Rozov et al., 2001) – decreasing use-dependent facilitation. It is possible that the use of EGTA in the internal solution has flattened the disparity between the use-dependent dynamics of in each layer, and also may have implications for comparison across species. This is a major problem that requires addressing.

We agree that the internal solution buffer concentration selected for the integration test will impact the short-term dynamics. We’ve now emphasized this point in serval places in the manuscript.

We also conducted additional experiments characterizing STP without EGTA in the intracellular solution in layer 5 mouse lines. We selected the layer 5 mouse lines because the STP in 0.3 mM EGTA differed from previous work in layer 5 (Reyes and Sakmann, 1999, Lefort and Petersen, 2017). We’ve included this data in a new supplementary figure (Figure 5—figure supplement 1) and added associated discussion.

To facilitate comparisons across pipeline datasets and for logistical considerations we sought to minimize the differences with existing cell-types pipelines (http://celltypes.brain-map.org/). Since the cell-types pipeline (http://celltypes.brain-map.org/data) had already begun production with 0.3 mM EGTA, we selected the same concentration for the integration test. Nevertheless, because we share the reviewer's concerns, we will characterize the STP without EGTA in the internal solution during production pipeline experiments.

4) It has been shown that connectivity depends on the pre- and post-synaptic cell type (see, for example, Brown and Hestrin. 2009). However, here the authors are likely mixing synapse types in their analyses. Cre is expressed in more than one cell type in the Ntsr1-Cre line (Chevée et al., 2018, Tasic et al., 2016) and the Tlx3-Cre line (Tasic et a.l, 2017). Layer 2/3 includes more than one cell type (see, for example, Yamashita et al., 2013; Yamashita et al., 2018; Tasic et al., 2016; Tasic et al., 2017). Thus, the connectivity rates and the average synaptic properties shown here may not represent connectivity rates among specific cell types but rather represent mixtures of synapse types within a layer.

We have amended the Introduction to emphasize that we are assessing connectivity among cell classes (groups of related cell types). We also corrected an accidental use of the term `cell-type` when we should have written `cell class` (Discussion section). Our experiments have a similar resolution to previous reports (e.g., Brown and Hestrin’s labeled classes equate roughly to our layer 5 Cre populations in terms of class size). We’ve added a section to the discussion addressing the implications of assessing connectivity at the level of cell class.

The work we present here is part of larger framework: A goal of the Allen Institute for Brain Science is to use physiological and anatomical data modalities, including connectivity and connection dynamics, to evaluate their association with transcriptomically defined cell types. Where we observe diversity in the synaptic dynamics within a class, we will then follow up with experiments that allow us to identify if the different dynamics are associated with different transcriptomically defined cell types. We adopted our current strategy to reduce the number of synapse types we must explore with (transcriptomic) cell type resolution.

5) The machine learning approach is basically a form of advanced template matching that looks for an event with the right properties. But at no stage can we see that the machine-learning algorithm knows exactly where the presynaptic spike is. Detection should explicitly model the fact that the time of the action potential is known, since it would strengthen its performance, in turn affecting estimates.

We have clarified the Materials and methods section to indicate that 3 ms window over which responses are analyzed is placed 1 ms after the rising phase of the presynaptic action potential (see also Figure 1—figure supplement 1B). Peak amplitudes and latencies measured from this window provide reasonably effective features for classification, especially when compared to the same values measured from background noise. However, we find that typical latency jitter is on the order of 100 μs, so we agree that there is potential here to improve classification accuracy by further constraining the response window. At the same time, this introduces the possibility of entirely missing connections if the initial latency estimate is inaccurate.

6) Figure 4EF data set is small, and the resulting connectivity rate in panel F is low compared to the prior literature (e.g. Perin, Berger and Markram, 2011). Either this n should be increased, or maybe simply remove panels E-F?

We have increased the sample size of the 2-photon experiments used to probe recurrent connectivity among Tlx3 neurons at distances greater than can be feasibly examined with multipatch. The enhanced sample size improves the confidence in the connectivity estimate.

To our knowledge, previous applications of 2P optogenetics for circuit mapping were proof of principle in nature. We believe this is one of the first biological results. Furthermore, this is the first corroboration of the 2P optogenetic circuit mapping with multipatch data.

7) What about bidirectional connections? Both Song, 2005 and Lefort, 2009 make a big deal of those, it would seem strange not to mention them.

A major limitation of measuring reciprocal connectivity on heterogeneous populations of cells is that it can yield results that appear to differ from random chance even if they do not (Hoffmann and Triesch, 2017). Furthermore, very high sampling is required to confidently measure differences in the reciprocal connectivity of sparsely connected cells.

We’ve added our reasoning for not including reciprocal analysis to the manuscript.

8) E.g. albino animals are known to have different connectivity patterns, so if some of these transgenic lines (Materials and methods section, etc.) were created on an albino background, then please discuss. Differences attributed to layers could in reality be due to the transgenic mouse line used.

All mice in this study are on a C57BL/6J background and are congenic C57BL/6J (at least six generations bred to C57BL/6J). By making all our mice congenic, we can be sure that any phenotype that does arise is a result of the various drivers/reporters of the mouse, rather than its genetic background. We’ve updated the Materials and methods section.